# An inhibitory segment within G-patch activators tunes Prp43-ATPase activity during ribosome assembly

Daniela Portugal-Calisto[1], Alexander Gregor Geiger [1], Julius Rabl[2], Oscar Vadas [3], Michaela Oborská-Oplová [1], Jarosław Mazur [1], Federica Richina[4], Purnima Klingauf-Nerurkar[1,4], Erich Michel [5], Alexander Leitner [6], Daniel Boehringer [2] & Vikram Govind Panse [1,7] ✉

Mechanisms by which G-patch activators tune the processive multi-tasking ATP-dependent RNA helicase Prp43 (DHX15 in humans) to productively remodel diverse RNA:protein complexes remain elusive. Here, a comparative study between a herein and previously characterized activators, Tma23 and Pxr1, respectively, defines segments that organize Prp43 function during ribosome assembly. In addition to the activating G-patch, we discover an inhibitory segment within Tma23 and Pxr1, I-patch, that restrains Prp43 ATPase activity. Cryo-electron microscopy and hydrogen-deuterium exchange mass spectrometry show how I-patch binds to the catalytic RecA-like domains to allosterically inhibit Prp43 ATPase activity. Tma23 and Pxr1 contain dimerization segments that organize Prp43 into higher-order complexes. We posit that Prp43 function at discrete locations on pre-ribosomal RNA is coordinated through toggling interactions with G-patch and I-patch segments. This could guarantee measured and timely Prp43 activation, enabling precise control over multiple RNA remodelling events occurring concurrently during ribosome formation.

ATP-dependent RNA helicases are involved in different aspects of gene expression including mRNA splicing and nuclear export, pre-ribosomal RNA (pre-rRNA) folding, and turnover[1-3]. By melting secondary structure elements, members of the processive DEAH/RHA-box RNA helicase family promote strand separation or annealing to remodel dynamic RNA:protein complexes[4,5]. The irreversible nature of these events necessitates mechanisms that tightly regulate the timing of their processive activities.

Prp43 (human DHX15) is an archetypical multitasking DEAH/RHA-box RNA helicase which functions during pre-mRNA splicing, spliceosome complex disassembly, ribosome biogenesis, cap-proximal mRNA methylation, and host defence during viral infections[6]. It contains two RecA-like domains connected by a loop, a C-terminal winged-helix (WH) followed by the Ratchet, and an oligosaccharide-binding (OB) domains[7-9]. A cleft formed between the two RecA-like domains bears the motifs responsible for ATP binding and hydrolysis. Contacts between the C-terminal domains and the RecA-like domains regulate the ability of the RNA-binding tunnel to interact with RNA. Structural studies have provided insights into how the RecA-like domains couple RNA unwinding activity with ATP hydrolysis[10]. During processive 3′–5′ RNA unwinding, the RNA-binding tunnel accommodates a single RNA strand[11]. This interaction is RNA-sequence independent and involves the sugar–phosphate backbone[11]. The specificity of Prp43 to RNA substrates presumably is provided by

[1]Institute of Medical Microbiology, University of Zurich, Zurich, Switzerland. [2]Cryo-EM Knowledge Hub, ETH Zurich, Zurich, Switzerland. [3]Faculty of Medicine, University of Geneva, Geneva, Switzerland. [4]Institute of Biochemistry, ETH Zurich, Zurich, Switzerland. [5]Department of Biochemistry, University of Zurich, Zurich, Switzerland. [6]Institute of Molecular Systems Biology, ETH Zurich, Zurich, Switzerland. [7]Faculty of Science, University of Zurich, Zurich, Switzerland. ✉e-mail: vpanse@imm.uzh.ch

members of the G-patch protein family[12–15]. In addition, G-patch cofactors boost Prp43-ATPase activity locally to drive processive RNA unwinding/annealing events[15–17]. Yet, how Prp43-activation is controlled at its functional location to avoid unproductive premature remodelling remains enigmatic.

Members of the G-patch family of proteins are characterized by a glycine-rich motif of ~40-50 amino acids[18]. Bioinformatic analyses have identified >20 G-patch proteins in humans, however, many of their target RNA helicases remain unresolved[12,18]. Four G-patch proteins in yeast (Ntr1/Spp382, Pfa1/Sqs1, Pxr1/Gno1 and Cmg1) have been shown to activate Prp43-ATPase activity. The nuclear localized Ntr1 recruits Prp43 to the spliceosome to disassemble intron-lariat complexes and stalled spliceosome complexes associated with aberrant splicing events[12,19,20]. Together with the G-patch cofactor Pfa1 (NKRF in humans), Prp43 activity has been proposed to prepare 20S pre-rRNA for processing into mature 18S rRNA[21,22]. Functional studies indicate the requirement of the G-patch cofactor Pxr1 (PINX1 in humans) for early nucleolar pre-rRNA processing steps and a separate role in Rrp6-dependent 3′-end processing of small nucleolar RNAs (snoRNAs)[16,23,24]. Prp43 also interacts with the cytoplasmic and mitochondrial G-patch cofactor Cmg1, but the function of this complex remains unknown[15].

Structural and biochemical studies show that the G-patch of NKRF makes at least two critical contacts with DHX15 to regulate its ATPase activity[25]. Specifically, a brace-helix at the beginning of the G-patch and a disordered brace-loop at the end of the G-patch contact the WH within the C-terminal domain and the RecA2 domain of the RNA helicase, respectively[25]. In this way, the G-patch tethers the two RecA-like domains of Prp43, promoting a closed RNA-binding tunnel conformation, which is thought to improve the processivity of the helicase by favouring RNA binding[25]. Single-molecule FRET studies demonstrated that the G-patch also opens the catalytic cleft of Prp43 to promote ADP-ATP exchange, and stimulate ATPase activity[26].

Prp43 co-enriches with 90S, 60S, and 40S pre-ribosomes which contain 35S, 27S and 20S pre-rRNAs, respectively[27]. Crosslinking and cDNA analysis (CRAC) data show that Prp43 binds to multiple sites within 18S and 25S regions of the pre-rRNA[22,28]. Curiously, the Prp43-binding sites within the 25S rRNA region were found to be in the proximity of target sequences of snoRNAs, which accumulate on 60S pre-ribosomes in catalytically dead Prp43 mutants or when Prp43 is depleted[22]. Based on these data, Prp43 activity has been implicated in the release of a specific set of snoRNAs and in the recruitment of a different set of snoRNAs[22,28].

Here, we report the discovery of a nucleolar localised G-patch-containing cofactor, Tma23 (Translation Machinery Associated 23), that is required during 60S assembly. In addition to the activating G-patch segment, we reveal an inhibitory segment within Tma23 that restrains Prp43-ATPase activity. We find this inhibitory segment in another G-patch cofactor, Pxr1, that also functions during 60S assembly. Tma23 and Pxr1 organize into homo-dimeric complexes suggesting that Prp43 function at multiple sites on pre-rRNA can be tuned and coordinated through toggling interactions with the activating G-patch and inhibitory segments.

## Results

### Tma23 is a G-patch activator of Prp43

To discover G-patch-containing factors that regulate Prp43-ATPase activity, we employed a tandem affinity purification (TAP) approach in the model organism budding yeast[29]. We integrated a C-terminal TAP tag at the *PRP43* gene *locus* to generate a yeast strain expressing a Prp43-TAP fusion protein. The resulting strain grew at wild-type rates suggesting that the TAP tag did not interfere with the essential function(s) of Prp43 (Supplementary Fig. 1a). Prp43-TAP was affinity purified (Fig. 1a, left panel) and the co-enriching proteome was analysed by label-free quantitative mass spectrometry (Fig. 1a, middle panel). 477 significantly co-enriching proteins were clustered based on GO

analysis using the DAVID tool[30]. As expected, the Prp43-TAP proteome co-enriched factors required for pre-mRNA splicing, pre-mRNA processing, pre-rRNA maturation and ribosome assembly (Fig. 1a, right panel)[14,22,27,31,32]. Prp43-TAP co-enriched known G-patch activators: Ntr1, Pfa1, and the nucleolar localized Pxr1 (Fig. 1a, middle panel; and Supplementary Fig. 1b)[14,16,17,23]. Although Prp43-TAP co-enriched components involved in mitochondrial translation (Fig. 1a, right panel), the mitochondrial localized G-patch cofactor Cmg1[15] was not detected.

Close inspection of protein sequences of known G-patches indicated two features: a -G(WY)KXG- motif located at the C-terminal end of the characteristic brace-helix, and a downstream -G(LI)G- motif forming the characteristic brace-loop region (Fig. 1b and Supplementary Fig. 1c). Mining the Prp43-TAP enriched proteome for these features, revealed a nucleolar localized protein, Tma23 (Fig. 1a, middle panel; 1b, and Supplementary Fig. 1b). Tma23 was identified in proteomic analyses of ribosomal complexes[33–35]. An unbiased genetic screen revealed a functional interaction between Tma23 and another nucleolar factor Emg1, which is associated with the 40S subunit precursor, the 90S pre-ribosome[36–38]. However, the contribution of Tma23 to the process of ribosome assembly remained unexplored.

We initiated our study by validating the interaction between Tma23 and Prp43 in vivo. For this, we constructed yeast strains expressing Tma23-TAP and Pxr1-TAP (positive control) by integrating a C-terminal TAP tag at their respective genomic *loci*. The resultant strains grew at wild-type rates (Supplementary Fig. 1a). Tma23-TAP and Pxr1-TAP were affinity purified, and Prp43 co-enrichment was assessed by Western blotting. Tma23-TAP and Pxr1-TAP co-enriched Prp43, thus validating the proteomic data (Fig. 1c).

Next, we dissected the interaction between the G-patch activators and Prp43 using yeast two-hybrid (Y2H) and in vitro pulldown assays. Y2H studies show that Tma23 and Pxr1 bind to Prp43 as judged by growth on media lacking histidine (SD-His) (Fig. 2a, rows 1 and 6). Truncation studies show that Prp43 interacts with the middle domain of Tma23 and Pxr1 (Tma23^M and Pxr1^M; Fig. 2a, rows 4 and 9), respectively. These findings were validated by in vitro pulldown assays (Fig. 2b; lanes 3 and 9). Pxr1^M and Tma23^M are degradation-prone when expressed alone in *E. coli*. However, when co-expressed with Prp43, these fragments form stable stochiometric Prp43:Pxr1^M and Prp43:Tma23^M complexes (Fig. 2b; lanes 3 and 9).

We did not detect stable interactions between Prp43 and the C-terminal segments (Dimerization segment, see later) of Pxr1 and Tma23 (Pxr1^D and Tma23^D) using Y2H and in vitro binding assays (Fig. 2a, rows 5 and 10; Fig. 2b, lanes 4 and 10). Y2H studies also did not reveal interactions between Prp43 and the G-patch segments (Tma23^G and Pxr1^G) (Fig. 2a, rows 3 and 8). While the G-patch of Pfa1 (Pfa1^G) formed a stable complex with Prp43 in vitro (Fig. 2b, lanes 5 and 11), the G-patches of Tma23 and Pxr1 (GST-Tma23^G and GST-Pxr1^G) did not (Fig. 2b, lanes 2 and 8). AlphaFold2-derived models[39] suggest that the GST-Tma23^G, GST-Pxr1^G, and GST-Pfa1^G (positive control) interact with Prp43 in a canonical manner (Supplementary Fig. 1d). We infer that Pxr1^G, and presumably Tma23^G, engage transiently with Prp43 to stimulate its ATPase activity.

We assessed the stimulation of Prp43-ATPase activity by Tma23^G using an established enzyme-coupled assay[40] based on the ATP-dependent conversion of phosphoenolpyruvate to pyruvate, by pyruvate kinase, and then to lactate, by lactate dehydrogenase. The latter step requires NADH, whose oxidation is proportional to the rate of ATP hydrolysis and can be monitored by the reduction in absorption at 340 nm over time. Pfa1^G and Pxr1^G were used as positive controls for these analyses. We found that Prp43-ATPase activity was strongly stimulated when in a 1:1 complex with Pfa1^G (Fig. 2c, column 5). In comparison, Tma23^G is a weaker activator of Prp43-ATPase activity since the addition of a 10-fold molar excess of Tma23^G resulted only in a 3-fold stimulation (Fig. 2c, compare columns 2, 3 and 4). Pxr1^G, even in

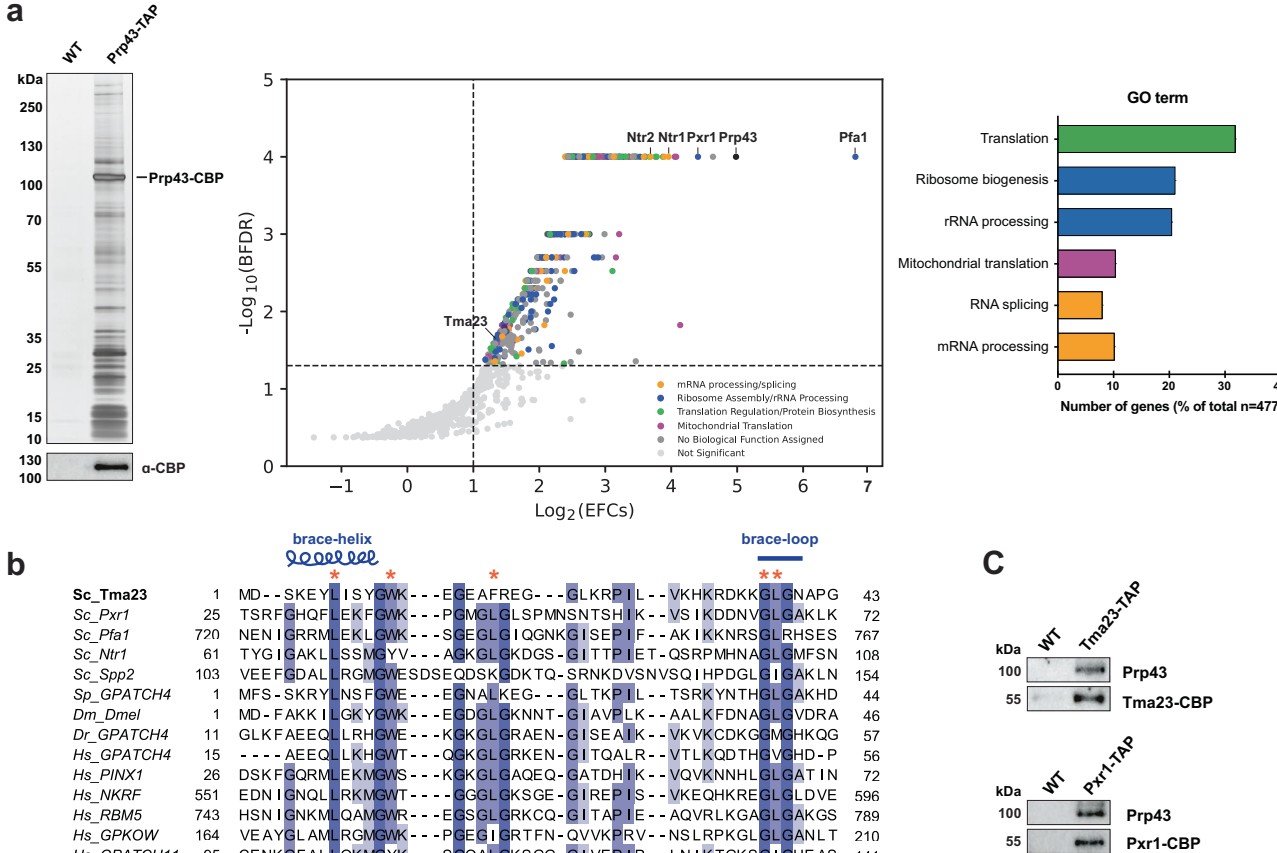

**Fig. 1 | Tma23 is a G-patch containing Prp43 activator. a** Left panel: Prp43-TAP eluate was separated on a NuPAGE 4–12% Bis-Tris gradient gel and analysed by Silver staining or Western blotting using α-CBP antibody. WT (BY4741 untagged strain) was used as negative control. Middle panel: Prp43-TAP proteome was identified by label-free quantitative mass spectrometry analysis. The data was analysed with the R package prolfqua[58]. The plot shows a Bayesian false discovery rate (BFDR) of less than 10% as a function of empirical fold change (EFC) score >2. Right panel: Statistically significant proteins (*p*-value < 0.001) were grouped based on the DAVID functional GO clustering analysis. This experiment was performed independently three times with similar results. **b** Sequence alignments of Tma23 with known G-patch proteins across species: (Sc) *Saccharomyces cerevisiae*; (Sp) *Schizosaccharomyces pombe*; (Dm) *Drosophila melanogaster*; (Dr) *Danio rerio*; (Hs) *Homo sapiens*. Brace-helix and brace-loop as defined previously[25] are indicated on top. The alignments were made with MAFFT[73] and visualized with Jalview (version 2.11.2.5)[74–76]. Asterisks indicate residues substituted for functional studies. **c**, Tma23-TAP and Pxr1-TAP eluates were separated on a NuPAGE 4-12% Bis-Tris gradient gel and subjected to Western blotting using antibodies directed against Tma23, Pxr1, and Prp43. This experiment was performed independently three times with similar results. Source data are provided as a Source Data file.

10-fold molar excess, also weakly stimulated Prp43-ATPase activity (Fig. 2d, compare columns 2, 3 and 4).

Pfa1[G] forms a stable complex with Prp43 and strongly stimulates its ATPase-activity. We suspected that low-affinity between Prp43 and the two G-patches (Tma23[G] and Pxr1[G]) could underlie the weak stimulation. Hence, we generated two fusion proteins (Prp43-T[G] and Prp43-P[G]) wherein the C-terminus of Prp43 was fused via a linker to the N-terminus of Tma23[G] and Pxr1[G]. AlphaFold2-derived models suggest that Tma23[G] and Pxr1[G], in context of the fusions, can interact with Prp43 like a canonical G-patch (Supplementary Fig. 1e). Consistent with these in silico studies, the Prp43-T[G] fusion showed an ~30-fold stimulation as compared to Prp43 activity measured with the addition of equimolar amount of Tma23[G] (Fig. 2e, compare columns 3 and 4). In contrast, the Prp43-P[G] fusion showed a weaker ~2-fold stimulation (Fig. 2f, compare columns 3 and 4). Based on all these data, we include Tma23 into the family of G-patch-containing proteins that stimulate Prp43-ATPase activity.

**Tma23[G] and Pxr1[G] contribute to the 60S assembly pathway**

To assess the role of Tma23 in vivo, we generated a conditional yeast mutant in which the endogenous TMA23 was placed under the control of a galactose-inducible promoter (P$_{GAL1}$-*TMA23*). On repressive glucose-containing media, the P$_{GAL1}$-*TMA23* strain was severely growth impaired (Fig. 3a). Likewise, the P$_{GAL1}$-*PXR1* strain was severely growth impaired on glucose-containing media (Fig. 3a). These data show that *TMA23*, like *PXR1*, is not an essential gene, but is required for optimal growth.

We localized the established 40S reporter (uS5-GFP) and 60S reporter (uL18-GFP) in Tma23- and Pxr1-depleted cells. As controls, we monitored the location of these reporters in *bud20Δ* and *yrb2Δ* cells, which are impaired in 60S and 40S pre-ribosome export, respectively (Fig. 3b, right panel)[41]. We found that uL18-GFP, but not uS5-GFP, accumulated in the nucleoplasm of Tma23 and Pxr1-depleted cells (Fig. 3b, left and middle panels) indicating a role for Tma23 and Pxr1 in the assembly of export competent 60S subunits. To measure the contribution of Prp43 activation mediated by Tma23 and Pxr1 in vivo, we generated a battery of G-patch point mutants (Fig. 3c). Y2H and in vitro binding studies showed that these substitutions did not impair the interaction between the mutant proteins and Prp43 (Supplementary Fig. 2a and 2b). Tma23[GM] / Pxr1[GM] and their different variants when expressed alone in *E. coli* are degradation prone. However, when co-expressed with Prp43 they form stable stochiometric complexes (Supplementary Fig. 2b). Strains expressing Tma23[L7E] and Pxr1[L33E] point mutants were growth impaired as judged by the size of individual

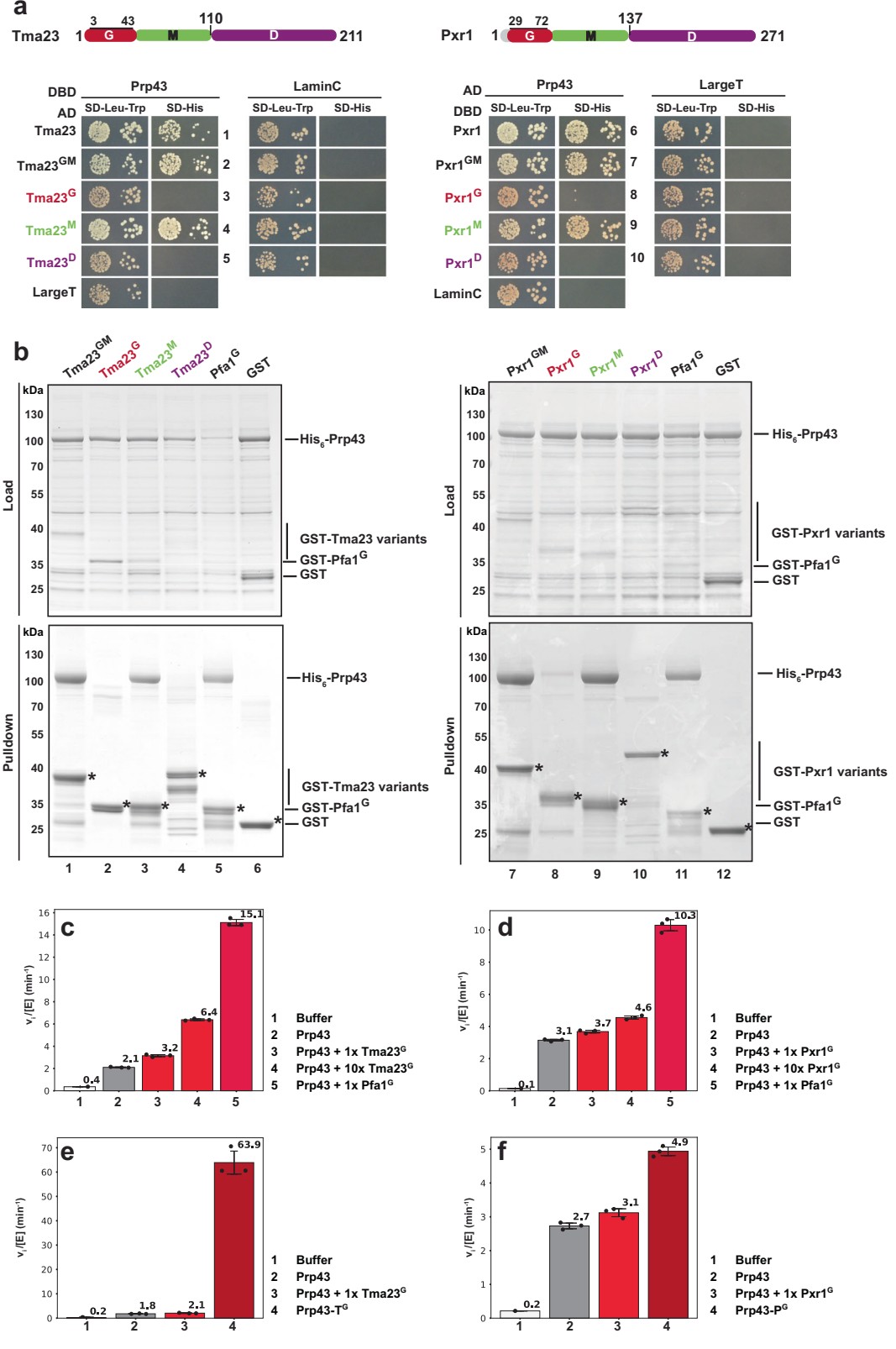

colonies (Fig. 3d, rows 3 and 6). These residues within Tma23 (L7E) and Pxr1 (L33E) are located within a conserved brace-helix, a segment important for stimulating Prp43-ATPase activity[25]. Like Tma23- and Pxr1-depleted cells, the Tma23[L7E] and Pxr1[L33E] mutants exhibit a strong nuclear accumulation of the 60S reporter, uL18-GFP (Fig. 3e). We suggest that Prp43 activation mediated by Tma23[G] and Pxr1[G] is required for efficient 60S assembly.

## Tma23[M] and Pxr1[M] restrain Prp43-ATPase activity

Y2H and biochemical studies showed that Tma23[M], but not Tma23[G], forms a stable complex with Prp43 (Fig. 2a, row 4 and 2b, lane 3). We wondered whether Tma23[M] provides a binding platform to bring Prp43 in the vicinity of Tma23[G] to bolster its ATPase activity. Surprisingly, the stimulation of Prp43-ATPase activity in a 1:1 complex with Tma23[GM] was weaker than the activity in the presence of a 10-fold molar excess of

**Fig. 2 | Tma23$^M$ and Pxr1$^M$ mediates binding to Prp43. a** Upper panel: Domain organization of Tma23 and Pxr1. Lower panel: Plasmids encoding indicated *LexA* DNA-binding domain (DBD) and *GAL4* activation domain (AD) fusion proteins were transformed into NMY32. Transformants were spotted in serial 10-fold dilutions on the indicated selective media and incubated at 30 °C for 2–4 days. LaminC and LargeT served as negative controls. **b** GST tagged variants of Tma23 and Pxr1 were co-expressed with His$_6$-Prp43 in *E. coli* cells and affinity-purified using Glutathione Sepharose 4 Fast Flow beads (Cytiva). The bound proteins were eluted using Pierce™ LDS Sample buffer at 70 °C, separated on a SurePAGE 4-20% BIS-TRIS gradient gel and visualized by Coomassie Blue staining. Pfa1 G-patch (Pfa1$^G$) and

GST-alone were used as positive and negative controls, respectively. Asterisks indicate the baits. **c**–**f** ATP hydrolysis rates at 2.5 mM ATP normalized for enzyme concentration (Prp43). Initial ATP hydrolysis rates were calculated by applying a linear regression to the NADH oxidation over time and normalized to the enzyme concentration for Prp43. Each independent experiment is shown (dots) and error bars indicate mean values ± SD. **c**–**d** Tma23$^G$ or Pxr1$^G$ was added to Prp43 in a 1:1 or 10:1 molar ratio, as indicated. Pfa1$^G$ was used as control. **e**–**f** Tma23$^G$ or Pxr1$^G$ were fused to Prp43 (Prp43-T$^G$ and Prp43-P$^G$, respectively). ATPase activities of the fusions were compared to Tma23$^G$ or Pxr1$^G$ added to Prp43 in a 1:1 ratio. Source data are provided as a Source Data file.

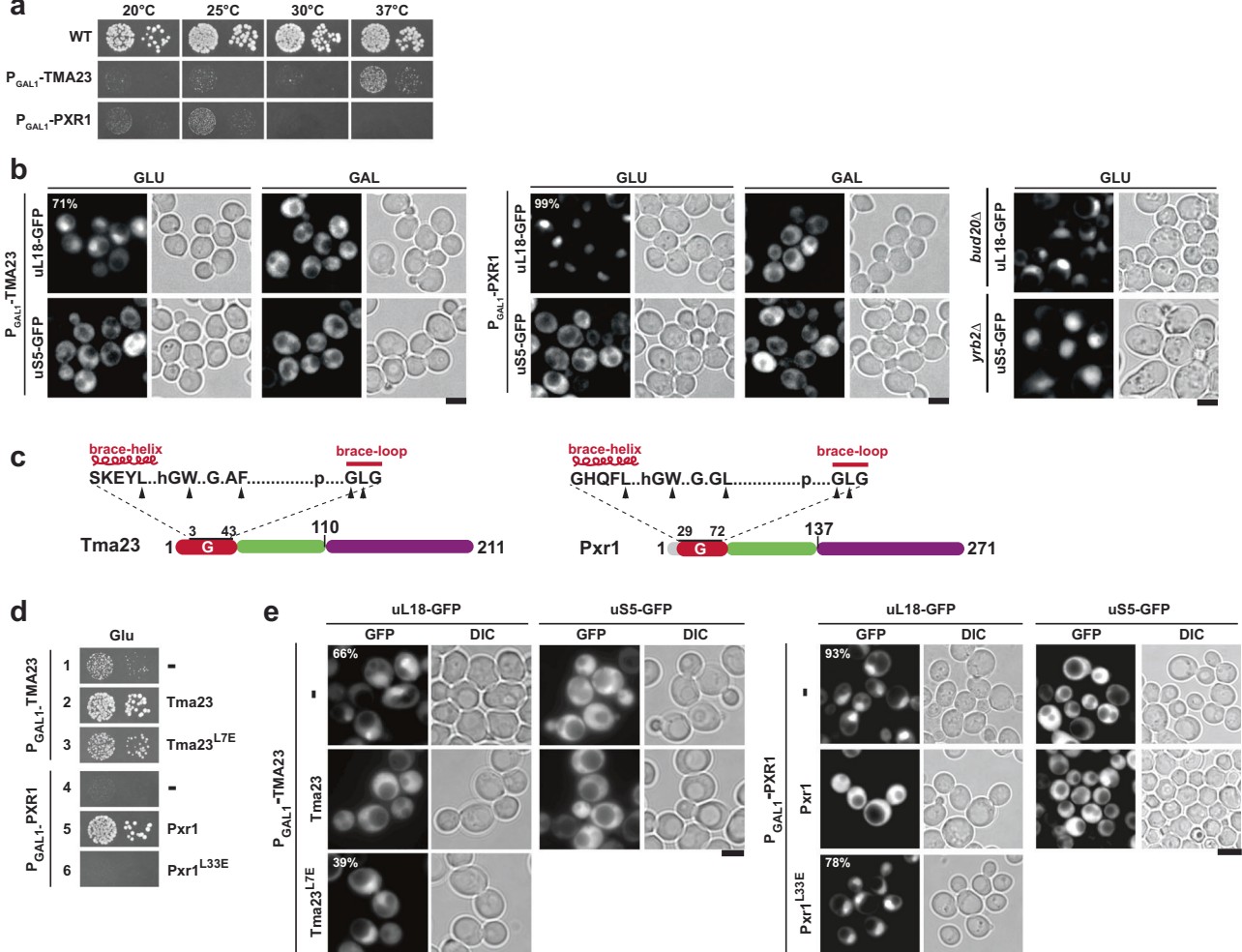

**Fig. 3 | Tma23$^G$ and Pxr1$^G$ mutants are impaired in 60S pre-ribosome export.** **a** Wild type (BY4741), P$_{GAL1}$-*TMA23*, and P$_{GAL1}$-*PXR1* strains were spotted in serial 10-fold dilutions on glucose-repressive medium and incubated at the indicated temperature for 3 – 7 days. **b** P$_{GAL1}$-*TMA23*, P$_{GAL1}$-*PXR1* expressing uL18-GFP or uS5-GFP, nuclear export reporters for 60S and 40S respectively, were grown to mid-log phase in glucose-(GLU) and galactose- (GAL) containing media at 37 °C. The location of the reporters was monitored by fluorescence microscopy and images were processed using ImageJ (version 1.50e; NIH and LOCI, USA). Scale bar = 5 µm. Percentage of cells exhibiting nuclear accumulation of uL18-GFP is indicated. *bud20Δ* and *yrb2Δ* were used as a positive control for nuclear accumulation of the reporters. This experiment was performed independently three times with similar

results. **c** Domain organization of Tma23 and Pxr1 showing partial sequence of the G-patch with mutated residues indicated by black arrowheads. **d** P$_{GAL1}$-*TMA23* strain expressing empty vector (-), Tma23 or Tma23$^{L7E}$ and P$_{GAL1}$-*PXR1* strain expressing empty vector (-), Pxr1, or Pxr1$^{L33E}$ were spotted in 10-fold serial dilutions on glucose-containing medium and incubated at 37 °C for 2–7 days. **e**, The indicated yeast strains expressing uL18-GFP or uS5-GFP reporters, were grown to mid-log phase in glucose-containing medium at 37 °C. Cellular localization of the reporters was visualized by fluorescence microscopy. The percentage of cells exhibiting nuclear accumulation of uL18-GFP reporter is indicated. This experiment was performed independently three times with similar results. Images were processed using ImageJ (version 1.50e; NIH and LOCI, USA). Scale bar = 5 µm.

Tma23$^G$ (Fig. 4a, compare columns 3 and 4). Functional studies showed that the L7E substitution within Tma23 induced a slow growth phenotype and a 60S export defect (Fig. 3d, e). Consistent with a critical role for the conserved leucine in Prp43 activation[25], we found that the ATPase activity of the Prp43:Tma23$^{GM-L7E}$ complex is lower as compared to the Prp43:Tma23$^{GM}$ complex (Fig. 4a, compare columns 4 and 5).

Interestingly, the ATPase activity of the Prp43:Tma23$^{GM-L7E}$ complex was found to be even lower than that of Prp43 alone (Fig. 4a, compare columns 2 and 5). These data hinted at a role for Tma23$^M$ in inhibiting Prp43-ATPase activity. In support of this idea, we found that a 1:1 complex of Tma23$^M$ with Prp43 inhibited the stimulation of Prp43-ATPase activity (Fig. 4a, compare columns 2 and 6). To obtain an

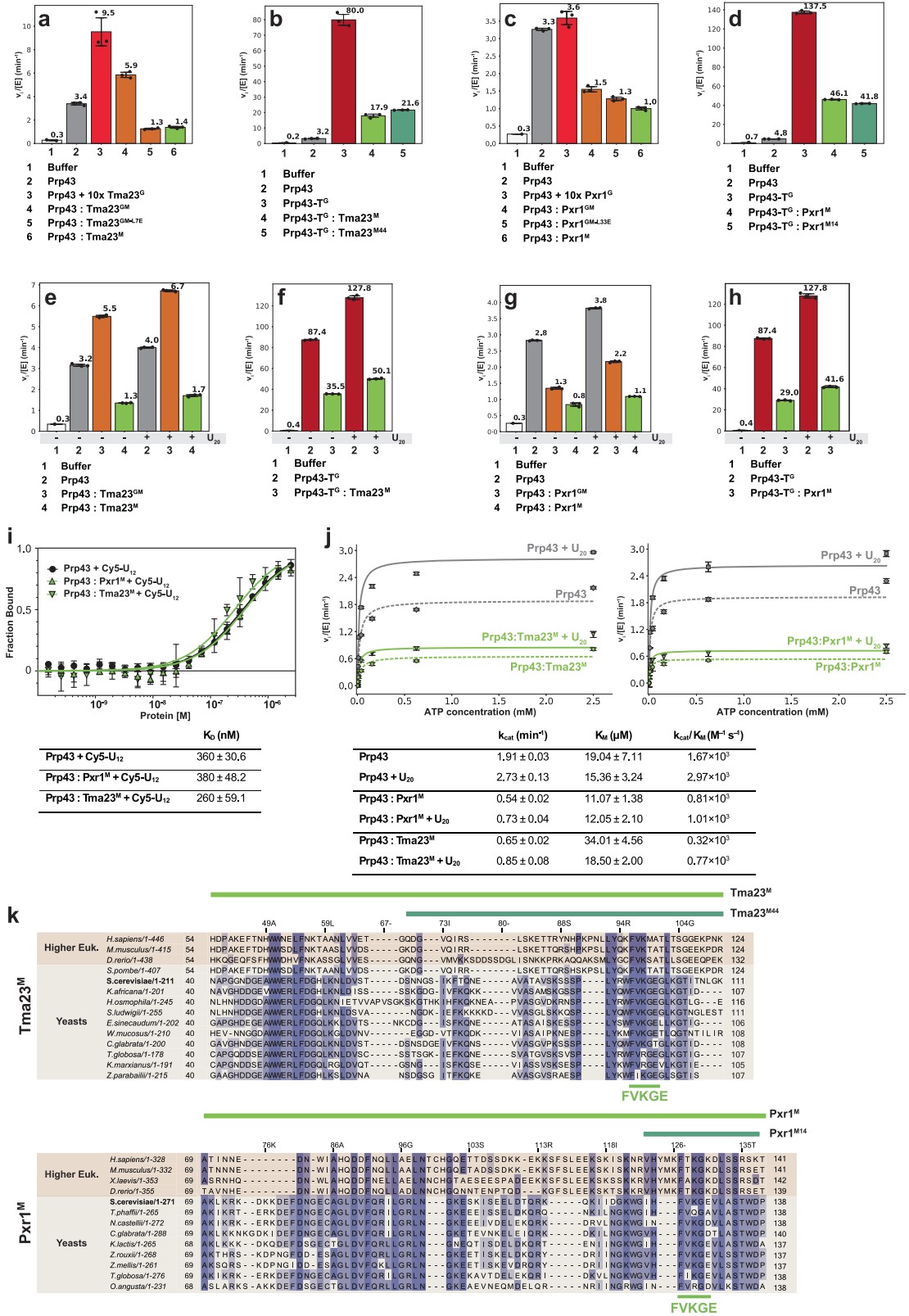

additional line of evidence for the inhibitory nature of Tma23[M], we compared the ATPase activities of Prp43-T[G] alone and in complex with Tma23[M]. The ATPase activity of the hyper-stimulated Prp43-T[G] fusion was strongly restrained when in complex with Tma23[M] (Fig. 4b, compare columns 3 and 4). A similar pattern of inhibition was observed for the Prp43:Pxr1[GM], Prp43:Pxr1[GM-L33E], Prp43:Pxr1[M] and Prp43-T[G]:Pxr1[M] complexes (Fig. 4c, d). We conclude that Tma23[M] and Pxr1[M] restrain Prp43-ATPase activity.

We investigated whether poly-$U_{20}$ RNA ($U_{20}$) can reverse the inhibitory hold of Tma23[M] and Pxr1[M] to stimulate Prp43-ATPase activity. As expected, $U_{20}$ stimulates the ATPase activity of Prp43 alone (Fig. 4e, compare grey columns, -/+RNA), the Prp43:Tma23[GM] complex (Fig. 4e, compare orange columns, -/+ RNA), and the Prp43-T[G] fusion (Fig. 4f, compare red columns, -/+RNA). However, the addition of $U_{20}$ is unable to reverse Tma23[M]-mediated inhibition of Prp43-ATPase activity (Fig. 4e, compare grey and green columns 2 and 4, -/+RNA).

**Fig. 4 | I-Patch restrains Prp43-ATPase activity. a–h** Initial hydrolysis rates at 2.5 mM ATP were normalized to Prp43 concentration. Initial rates were calculated by applying a linear regression to the NADH oxidation over time and normalized to Prp43 concentration. Each independent experiment is shown (dots) and error bars indicate mean values ± SD. **a** Tma23$^G$ was added to Prp43 in a 10:1 molar ratio. Tma23$^{GM}$, Tma23$^{GM-L7E}$, and Tma23$^M$ were co-expressed with Prp43. **b** ATPase activity of Prp43-T$^G$ was compared with Prp43-T$^G$:Tma23$^M$ or Prp43-T$^G$:Tma23$^{M44}$. **c** Pxr1$^G$ was added to Prp43 in a 10:1 molar ratio. Pxr1$^{GM}$, Pxr1$^{GM-L33E}$, and Pxr1$^M$ were co-expressed with Prp43. **d** ATPase activity of Prp43-T$^G$ was compared to either Prp43-T$^G$:Pxr1$^M$ or Prp43-T$^G$:Pxr1$^{MI4}$. **e** ATPase activities of Prp43:Tma23$^{GM}$ and Prp43:Tma23$^M$ complexes in presence/absence of U$_{20}$ RNA. **f** ATPase activities of Prp43-T$^G$ and Prp43-T$^G$:Tma23$^M$, in presence/absence of U$_{20}$ RNA. **g** ATPase activity of Prp43:Pxr1$^{GM}$ and Prp43:Pxr1$^M$ complexes in presence/absence of U$_{20}$ RNA. Pxr1$^{GM}$ and Pxr1$^M$ were co-expressed with Prp43. **h** ATPase activity of Prp43-T$^G$ and Prp43-T$^G$:Pxr1$^M$, in presence and absence of U$_{20}$ RNA. **i** K$_D$ determination for Prp43 binding to Cy5-labelled U$_{12}$-RNA by fluorescence polarization. Cy5-labelled U$_{12}$-RNA was titrated with Prp43 (circles), Prp43:Pxr1$^M$ (triangle up) or Prp43:Tma23$^M$ (triangle down) and the change in polarization was fitted to a single-site binding model (solid lines). An average of four independent measurements is plotted, and error bars indicate mean ± SD. **j** Kinetic parameters for ATPase activities of Prp43 (square, diamond), Prp43:Tma23$^M$ (triangle down, circle; left panel) and Prp43:Pxr1$^M$ (triangle down, circle; right panel), in the presence and absence of U$_{20}$, respectively. This experiment was performed independently three times with similar results. A representative measurement in triplicate is presented, and error bars indicate mean values ± SD. **k** Sequence alignments of Tma23$^M$ and Pxr1$^M$ across yeasts and higher eukaryotes. Light green lines indicate the M domain of Tma23 (Tma23$^M$) and Pxr1 (Pxr1$^M$). Dark green lines mark the segments that inhibit Prp43-ATPase activity (I-patch; Tma23$^{M44}$, Pxr1$^{MI4}$). The conserved FVKGE motif is indicated. Alignments were made with MAFFT[73] and visualized with Jalview (version 2.11.2.5)[74–76]. Source data are provided as a Source Data file.

Similarly, the ATPase activity of the Prp43-T$^G$ fusion that is inhibited when bound to Tma23$^M$ (Fig. 4f, compare red and green columns 2 and 3, -RNA) is not reversed by the addition of U$_{20}$ (Fig. 4f, compare red and green columns 2 and 3, +RNA). Similar observations were made for Prp43, Prp43:Pxr1$^{GM}$, Prp43:Pxr1$^M$ and Prp43-T$^G$:Pxr1$^M$ complexes (Fig. 4g, h). Fluorescence anisotropy-based measurements show that the presence of Tma23$^M$ or Pxr1$^M$ on Prp43 did not alter its affinity towards Cy5-labelled poly-U$_{12}$ RNA (Fig. 4i). All these data indicate that RNA binding does not reverse the inhibitory hold of Tma23$^M$ and Pxr1$^M$ on Prp43-ATPase activity. We obtained kinetic parameters for the ATPase activities of Prp43, Prp43:Tma23$^M$ and Ppr43:Pxr1$^M$ complexes, in the presence and in the absence of U$_{20}$ (Fig. 4j). We find that Tma23$^M$ and Pxr1$^M$ negatively impacts the k$_{cat}$ of Prp43-ATPase activity, whereas the K$_M$ remains in a similar μM range (Fig. 4j). These data point to an allosteric non-competitive inhibition mode of Prp43-ATPase activity by Tma23$^M$ and Pxr1$^M$.

We sought to pinpoint the region within Tma23$^M$ and Pxr1$^M$ responsible for the inhibition. We found a 44-residue region within Tma23$^M$ (Tma23$^{M44}$, Fig. 4k, upper panel) that inhibited Prp43-ATPase activity to a similar extent (Fig. 4b, compare columns 3, 4 and 5). Shorter constructs of Tma23$^M$ were unstable, hence we were unable to define a minimal region within Tma23$^M$ that restrains Prp43-ATPase activity. In contrast, shorter truncations of Pxr1$^M$ were stable and amenable to perform ATPase-assays. Using these complexes, we define a 14-residue segment (Pxr1$^{MI4}$), which we term I-patch, that robustly inhibited Prp43-ATPase activity (Fig. 4k, lower panel; and 4d, compare columns 3, 4 and 5).

**Prp43:I-patch interactions revealed by cryo-EM studies**

We sought to reveal the molecular basis for the regulation of Prp43-ATPase activity. For this, we prepared grids for single particle cryo-EM studies of Prp43:Tma23$^{GM}$ and Prp43:Pxr1$^{GM}$ complexes in different conditions, including the presence and absence of different nucleotides (ADP and ATPγS). The Prp43:Tma23$^{GM}$ complex was disassembled during vitrification in different conditions tested. Therefore, we focused our efforts to resolve the structure of the Prp43:Pxr1$^{GM}$ complex.

A cryo-EM map of the Prp43:Pxr1$^{GM}$ complex at an overall resolution of 3.3 Å (Supplementary Fig. 3 and Supplementary Table 1) showed a defined density for Prp43 residues 52-718 and Pxr1 residues 122-136 (Fig. 5a). Although full-length Prp43 was used for assembling the Prp43:Pxr1$^{GM}$ complex, no density was discernible in the cryo-EM map for the N-terminal region of Prp43 (1-48). The density for residues 49-51 of Prp43 was weak, suggesting conformational flexibility of the N-terminus. No density for most of the N-terminus of Pxr1 (residues 1-121), which encompasses the G-patch (residues 29-72), was observed, supporting the idea that this region is highly flexible and only transiently interacts with Prp43 to stimulate its ATPase activity. Despite incubating the Prp43:Pxr1$^{GM}$ complex with the non-hydrolysable ATPγS

before vitrification, the resulting density map showed only ADP bound to Prp43 (Fig. 5d, lower panel).

In the Prp43:Pxr1$^{GM}$ complex, only the very C-terminal region of Pxr1$^M$ (residues 122-136) were clearly resolved in the cryo-EM maps (Fig. 5a). This segment encompasses the 14-residue I-patch (residues 124-137) within Pxr1$^M$ which inhibits Prp43-ATPase activity (Fig. 4c, d). We found that the I-patch is anchored onto the RecA2 domain through a set of hydrophobic contacts and extends towards the RecA1 domain (Fig. 5a, b). F126, within the -FKVGE- sequence (Fig. 4k, green lines), clusters with hydrophobic residues F437 and L439 within Prp43 on the RecA2 domain (Fig. 5b, panel 1). The I-patch (residues 128-132) adopts a β-strand that is anti-parallel to the β-strand of Prp43 (residues 274-278) and extends the β-sheet from the centre of RecA2 domain. This involves contacts between L132 of Pxr1 and V274/L276 of Prp43 (Fig. 5b, panel 2). Another cluster at the C-terminal end of the I-patch involves the indole ring of W136 and the side chains of R270 and Y272 of Prp43 (Fig. 5b, panel 3), present within the loop that connects RecA1 with the RecA2 domain. These analyses indicate that Prp43 employs distinct interaction platforms to bind to the I-patch and to the G-patch (compare Fig. 5a and Supplementary Fig. 1c).

The RecA-like domains of the Prp43:Pxr1$^{GM}$ complex exhibit a conformation like the ADP-bound yeast Prp43 (RMSD = 0.66 Å, PDB 2XAU), CDP-bound Prp43, DHX8, or DHX15 (Supplementary Fig. 4a)[7,42–44], but markedly different from the non-hydrolysable ATP-analogue bound Prp43 complexes. ADP is bound in the cleft formed between the RecA1 and RecA2 domains of the Prp43:Pxr1$^{GM}$ complex (Fig. 5d and Supplementary Fig. 3c). R159 and F357 within RecA1 contact the adenine base in a similar manner as in Prp43-ADP complexes[11] (Fig. 5d). W136 of Pxr1 contacts R270 and Y272 within the loop connecting the RecA1 and RecA2 domains, which lies adjacent to the active site (Fig. 5b, panel 3). We find that this connecting loop in the Prp43:Pxr1$^{GM}$ complex occupies an outward position that is observed in all known ADP-bound structures of Prp43/DHX15[7–10,25,43] and Prp2[44] (Supplementary Fig. 4b). In Prp43:ATP-analogue-bound structures, this loop adopts a different conformation due to the reorientation of the RecA domains and is located 7 Å closer to the catalytic centre (Fig. 5c). Interestingly, in the Prp43:Pxr1$^{GM}$ complex, R270 and the catalytically important R430 within Prp43 adopt alternative conformations, which have not been observed in Prp43-ADP complex structures (Fig. 5d, upper panel).

Numerous structures of Prp43 and other DEAH-RNA helicases (Prp2 and Prp22) show striking conformational variability in the positioning of the C-terminal domains (WH, OB-fold and Ratchet) with respect to the two RecA-like domains. In all these structures, the conformation of the RNA-binding tunnel that is sandwiched between the RecA-like domains and the C-terminal domains, ranges from an RNA-bound closed state to an RNA-free open state (Supplementary Fig. 4c). We found that the RNA-binding tunnel of the Prp43:Pxr1$^{GM}$ complex adopts an open conformation in contrast to the activated closed Prp43:G-patch complexes (DHX15:NKRF$^G$; PDB:6SH6[25]; Fig. 5e).

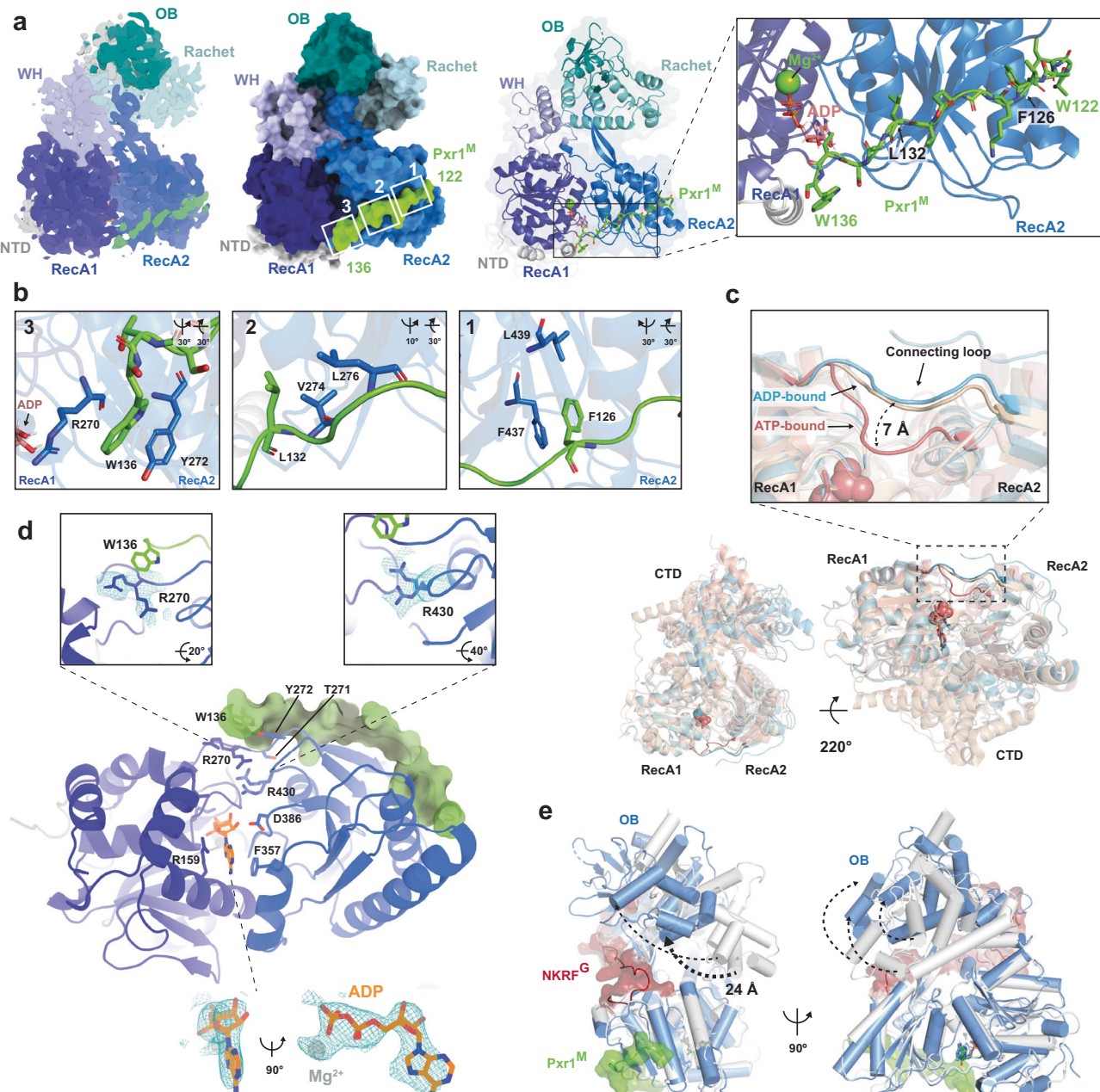

**Fig. 5 | Cryo-EM structure of a Prp43-Pxr1^GM complex. a** Cryo-EM structure of Prp43:Pxr1^GM complex at 3.3 Å resolution shows binding of Pxr1 peptide 122-136 to Prp43. Surface (left panel) and cartoon (right panel) representation of Prp43:Pxr1^GM complex. The individual domains of Prp43 are indicated and colour-coded in shades of blue. Pxr1 122-136 peptide is shown in green. Left panel: Pxr1^M peptide (green) contacts Prp43 RecA2 (blue) and the connecting loop between RecA2 and RecA1 (dark blue). The ADP and the magnesium ion are shown in pink/orange and green, respectively. **b** Hydrophobic contacts between Pxr1^GM and Prp43. Residues are depicted as sticks. **c** Prp43:Pxr1^GM complex, bound to ADP (blue), shows a 7 Å displacement of the connecting loop between RecA1 and RecA2 domains, super-posed with an ATP-bound Prp43 structure of *C. thermophilum* (light red, PDB

5LTA)[11]. DHX15:NKRF^G (beige) complex was used as representative of an ADP-bound structure (PDB 6SH6)[25]. **d** Upper panel: The density map of Prp43:Pxr1^GM complex shows two different conformations of Prp43 R430 and R270. When R430 is distant from the ADP, it contacts R270 and consequently, D386 does not interact with ADP (compared to other ADP-bound structures [PDB 2XAU][7]). Lower panel: The active site of Prp43 shows a density compatible with ADP. ADP is depicted in orange stick representation. **e** Structural comparison of Prp43:Pxr1^GM complex (blue and green) with DHX15:NKRF^G complex (grey and red; PDB 6SH6)[25] shows an open RNA-binding tunnel. The CTD of Prp43:Pxr1^GM complex is rotated outwards as compared to DHX15-NKRF^G complex by 24 Å as indicated by the arrows.

Interestingly, most of the particles of the Prp43:Pxr1^GM complex on the grids were found to be in the open conformation (Supplementary Fig. 3), which is consistent with a transient interaction of the G-patch with Prp43. In this conformation, the C-terminal Ratchet and the OB fold domains are rotated with respect to the RecA-like domains resulting in the repositioning of the OB fold domain by ~24 Å as compared to the G-patch-bound closed conformation (Fig. 5e).

To validate our structural studies, we focused on the well-resolved I-patch:Prp43 interaction. We substituted the conserved Phe within the I-patch of Pxr1^M (F126) and Tma23^M (F96) to Ala. The interaction of the Pxr1^M-F126A and Tma23^M-F96A mutant proteins with Prp43 was assessed by Y2H and in vitro pulldown assays. While Pxr1^M and Tma23^M formed a stable complex with Prp43 as judged by Y2H (Fig. 6a, rows 3 and 7) and in vitro pulldown assays (Fig. 6b,

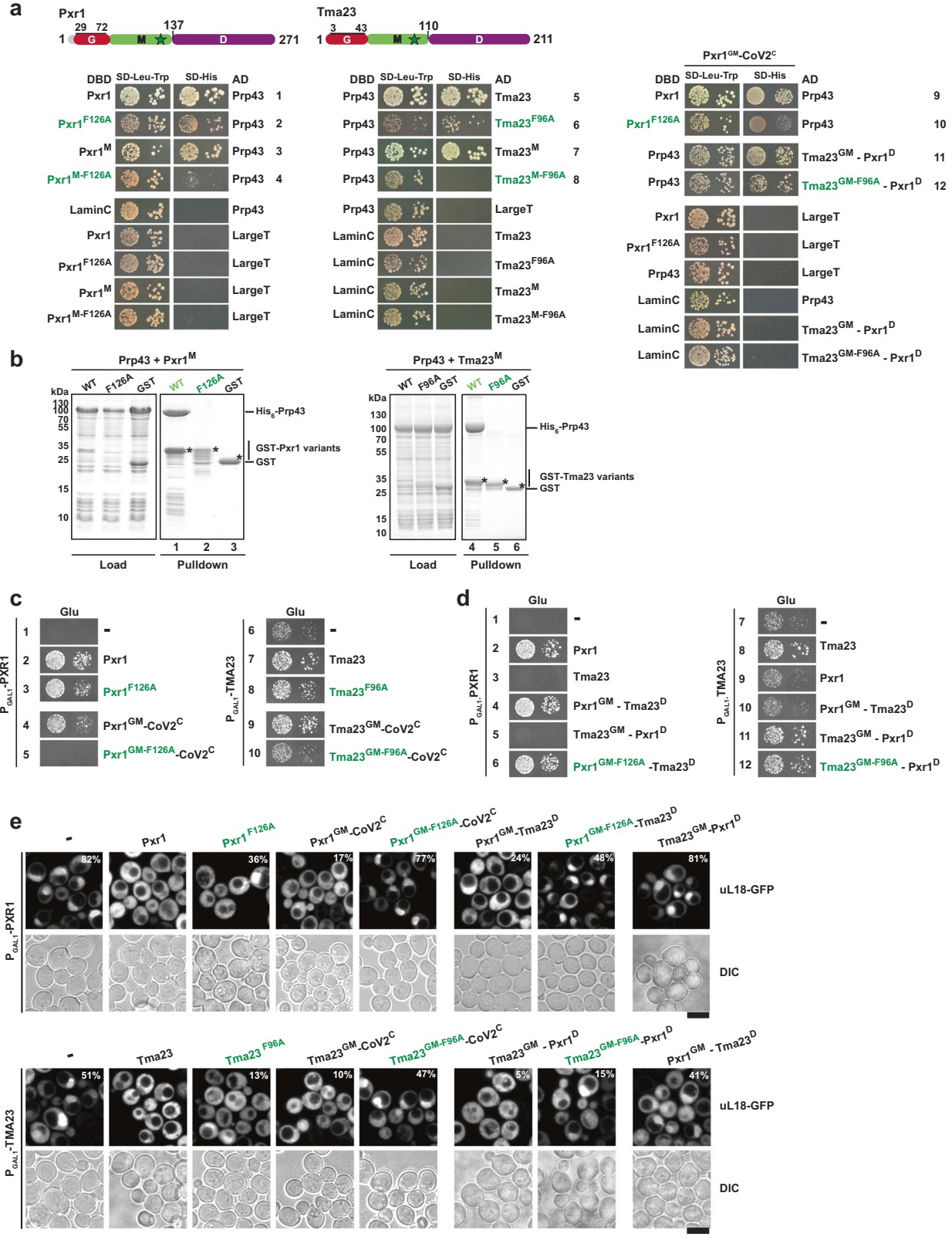

lanes 1 and 4), Pxr1^M-F126A and Tma23^M-F96A mutant proteins no longer bound to Prp43 (Fig. 6a, rows 4 and 8; and 6b, lanes 2 and 5). Y2H studies show that the Phe substitution in the context of the full-length Pxr1 and Tma23 (Pxr1^F126A and Tma23^F96A) did not abolish the interaction with Prp43 (Fig. 6a, rows 2 and 6). Yet, the Pxr1^F126A and Tma23^F96A point mutants did not fully complement the severe growth defect of Tma23- and Pxr1-depleted cells (Fig. 6c, rows 3 and 8).

These data implicate an additional surface employed by Tma23 and Pxr1 to bind to Prp43 (see later).

## HDX-MS analyses of Prp43:Pxr1^GM and Prp43:Tma23^GM complexes

Cryo-EM analyses of the Prp43:Pxr1^GM complex revealed how the I-patch binds to Prp43. However, these structural studies did not capture most

**Fig. 6 | A conserved Phe anchors I-patch to Prp43. a** Upper panel: Domain organization of Pxr1 and Tma23. The star indicates the substituted F126/F96 within the M-domains. Lower panel: NMY32 or a modified NMY32 Pxr1$^{GM}$-CoV2$^C$ strain transformed with plasmids encoding for the indicated *LexA* DNA-binding domain (DBD) and *GAL4* activation domain (AD) fusion proteins were spotted on selective SD-Leu-Trp and SD-His media and incubated at 30 °C for 2–4 days. LaminC and LargeT were used as negative controls. **b** Prp43 was co-expressed in *E. coli* with the indicated GST-fused wild-type proteins and mutants of Pxr1$^M$ and Tma23$^M$, or with GST, and affinity-purified using Glutathione Sepharose 4 Fast Flow beads (Cytiva). The bound proteins were eluted using Pierce™ LDS Sample buffer at 70 °C separated by SDS-PAGE and visualized by Coomassie Blue staining. GST was used as a negative control. Asterisks indicate the baits. This experiment was performed independently three times with similar results. **c** P$_{GAL1}$-*PXR1* and P$_{GAL1}$-*TMA23* strains expressing empty vector (-), the mutants Pxr1$^{F126A}$ and Tma23$^{F96A}$, or C-terminal

fusions of Tma23$^{GM}$ and Pxr1$^{GM}$ to CoV2 dimerization motif (CoV2$^C$) were transformed in the P$_{GAL1}$-*TMA23* and P$_{GAL1}$-*PXR1* strains were spotted in 10-fold serial dilutions on glucose-containing medium and incubated at 37 °C for 2–7 days. **d** P$_{GAL1}$-*TMA23* and P$_{GAL1}$-*PXR1* strains expressing empty vector (-), wild type Tma23 or Pxr1, or the indicated chimeric constructs were spotted in 10-fold serial dilutions on selective medium and incubated at 37 °C for 2–7 days. **e** P$_{GAL1}$-*PXR1* and P$_{GAL1}$-*TMA23* expressing the 60S nuclear export reporter uL18-GFP and empty vector (−), wild type protein or mutants (indicated) were grown to mid-log phase in glucose-containing media at 37 °C. The location of the reporter was analysed by fluorescence microscopy and images were processed using ImageJ (version 1.50e; NIH and LOCI, USA). The percentage of cells exhibiting nuclear accumulation of uL18-GFP reporter is indicated. This experiment was performed independently three times with similar results. Scale bar = 5 μm.

of Pxr1$^{GM}$. Of the 137 residues of Pxr1$^{GM}$, only 15 residues that encompass the I-patch, were resolved in these studies. The G-patch of Pxr1 was also not visible in the cryo-EM maps. Moreover, we were unable to perform cryo-EM studies of the Prp43:Tma23$^{GM}$ complex due to its dissociation during vitrification. We resorted to Hydrogen-Deuterium Exchange coupled to Mass Spectrometry (HDX-MS) and Crosslinking Mass Spectrometry (XL-MS) to gain insights into the organization and the dynamics of Prp43:Tma23$^{GM}$ and Prp43:Pxr1$^{GM}$ complexes.

HDX-MS tracks the exchange rates of protein amide hydrogens by mass spectrometry after exposure of proteins and/or protein complexes to D$_2$O[45]. Hence, it is a powerful tool to analyse protein dynamics and map interaction sites with ligands[46]. In addition to revealing direct interaction surfaces, HDX-MS also can pinpoint allosteric conformational changes associated with binding events. To map the interface of Prp43 with its partners in solution, we compared the hydrogen exchange rates of Prp43 alone and in complex with either Tma23$^{GM}$ or Pxr1$^{GM}$. Protein complexes were exposed to D$_2$O for 3, 30 and 300 s prior to quenching, pepsin digestion and analysis of peptide masses by MS. Global HDX-MS analysis of Prp43 showed that both N- and C-terminal extremities are highly flexible and poorly folded, with high D$_2$O incorporation levels within the first 50 and last 20 amino acids (Supplementary Fig. 5a and Supplementary Data 1). Comparison of HD exchange rates identified several key regions within RecA1 and RecA2 domains that were protected when in complex with either Pxr1$^{GM}$ or Tma23$^{GM}$ (Fig. 7a–c; Supplementary Fig. 5c). For example, the regions surrounding the catalytic centre of Prp43 are protected when bound to Tma23$^{GM}$ and Pxr1$^{GM}$ (Fig. 7a–c). Specifically, the α-helix harbouring the DEAH motif, important for ATP binding and hydrolysis[47], is protected (peptide 215-224). Another protected region, encompassing a β-strand–loop–β-strand, spans the surface of RecA1 and RecA2 domains and crosses over the entrance of the catalytic centre (peptide 259-276) (Fig. 7a–c). This protection can be readily explained as this region directly binds to the I-patch of Pxr1$^M$ (Fig. 7a). The RecA2 domain also shows extensive protection, suggesting that both Pxr1$^M$ and Tma23$^M$ binding seems to alter the rigidity of this domain as well as the functionally critical β-hairpin (peptide 391-411) that holds the 5′ end of the single-stranded RNA during processing[48]. Protection is also observed in the immediate vicinity of the brace-helix (peptide 477-491) and brace loop (peptide 440-448) binding sites of the G-patch (Fig. 7a–c; Supplementary Fig. 5b)[25]. Although Pxr1$^G$ is not visible in the cryo-EM maps of the Prp43:Pxr1$^{GM}$ complex, these protection data mapped on a AlphaFold2 model suggest that the interaction of the G-patch with Prp43 is likely to be canonical (Fig. 7a; Supplementary Fig. 5b). A similar protection was found in the same regions of the AlphaFold2 model of the Prp43:Tma23$^G$ complex indicating a similar interaction mode (Fig. 7a, Supplementary Fig. 5b). Overall, the HDX-MS analysis reveals a distinct protection pattern on the two RecA-like domains that correlate with the cryo-EM structure of the Prp43:Pxr1$^{GM}$ complex, and the AlphaFold2-model of the Prp43:Tma23$^G$ complex.

We employed XL-MS[49,50] as a complementary approach to obtain insights into the organization of the Prp43:Tma23$^{GM}$ complex. For these analyses, we employed the Lys-reactive crosslinking agent DSS. The reliable Lys-Lys crosslinks were mapped on a high-confidence AlphaFold2-generated model of a Prp43:Tma23$^M$ complex (Fig. 7d). The three crosslinks between Prp43 and Tma23$^{GM}$ indicate a proximity between the C-terminal region of Tma23$^M$ (residues 87-103) encompassing the I-patch and Prp43 RecA2 (residues 322-435) (Fig. 7d). We conclude that HDX-MS and XL-MS studies validate the cryo-EM structure of Prp43:Pxr1$^{GM}$ as well as support the AlphaFold2-model of the Prp43:Tma23$^M$ and Prp43:Tma23$^G$ complexes.

**C-terminal segments organize Pxr1 and Tma23 homodimers**

A previous study had suggested that Tma23 forms homodimers[36]. We investigated which region is responsible for Tma23 homodimerization by employing a Y2H approach. We found that Tma23$^D$, but not Tma23$^{GM}$, self-interact as judged by growth on SD-His plates supporting the notion that Tma23 homodimerizes via its C-terminal segment (Supplementary Fig. 6a). Likewise, Pxr1 homodimerizes via its C-terminal segment (Pxr1$^D$) (Supplementary Fig. 6b). To demonstrate that homodimerization is critical for Tma23 and Pxr1 function, we replaced Tma23$^D$ and Pxr1$^D$ with a heterologous nucleocapsid protein of SARS-CoV2 (CoV2$^C$) virus that forms a stable homodimer[51] (Supplementary Fig. 6c). We found that the fusions of Tma23$^{GM}$ and Pxr1$^{GM}$ with CoV2$^C$ (Tma23$^{GM}$-CoV2$^C$ and Pxr1$^{GM}$-CoV2$^C$) rescued the severe growth impairment of Tma23- and Pxr1-depleted cells, respectively, showing that Tma23$^D$ (Fig. 6c, compare rows 6, 7, and 9) and Pxr1$^D$ (Fig. 6c, compare rows 1, 2 and 4) function as dimerization segments. Consistent with these data, we found that Tma23$^D$ can be exchanged for Pxr1$^D$ and vice versa. The Pxr1$^{GM}$-Tma23$^D$ chimera significantly rescued the severe slow growth phenotype (Fig. 6d, compare rows 1, 2, and 4) and the 60S export defect (Fig. 6e, upper panel) associated with Pxr1-depleted cells. Similarly, the Tma23$^{GM}$-Pxr1$^D$ chimera rescued the severe slow growth phenotype (Fig. 6d, compare rows 7, 8, and 11) and 60S export defect associated with Tma23-depleted cells (Fig. 6e, lower panel). Tma23 and Tma23$^{GM}$-Pxr1$^D$ did not complement the severe growth (Fig. 6d, compare rows 1, 3, and 5) and the 60S export defect (Fig. 6e, upper panel) of Pxr1-depleted cells. Likewise, Pxr1 and Pxr1$^{GM}$-Tma23$^D$ also did not complement the growth (Fig. 6d, rows 7, 9, and 10) and 60S export defect (Fig. 6e, lower panel) of Tma23-depleted cells. These data show that Tma23$^{GM}$ and Pxr1$^{GM}$ cannot functionally replace each other.

Y2H studies showed that the interaction between Prp43 and Pxr1$^M$/Tma23$^M$ was abolished by substituting a specific Phe residue within the I-patch to Ala (Fig. 6a, left panel, compare rows 3 and 4; right panel, compare rows 7 and 8). But this was not the case in context of the full-length Tma23 and Pxr1 (Fig. 6a, left panel, compare rows 1 and 2; right panel, compare rows 5 and 6). To exclude that the Y2H interaction between Prp43 and Pxr1$^{F126A}$/Tma23$^{F96A}$ is not mediated by a bridging wild-type endogenous copy of Pxr1/Tma23, respectively, we

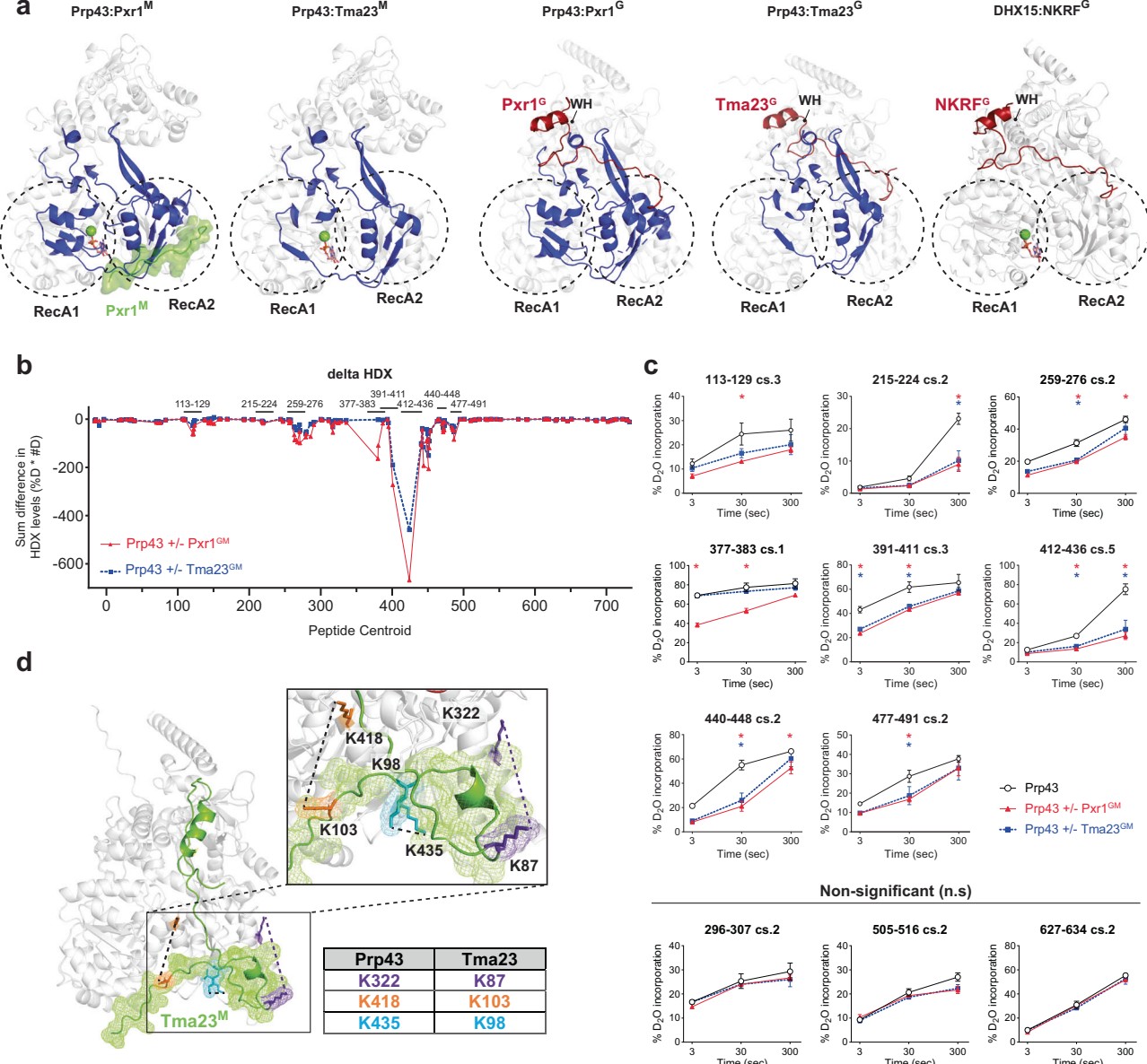

**Fig. 7 | HDX-MS and XL-MS of Pxr1$^{GM}$:Prp43 and Tma23$^{GM}$:Prp43 complexes.**
**a** Binding interface between Prp43 and either Pxr1$^{GM}$ or Tma23$^{GM}$ shows extensive protection of both RecA-like domains. Regions of Prp43 showing reduced H/D exchange rates upon complex formation are highlighted in blue. Regions in green indicate parts of Pxr1 present in the cryo-EM structure (left). The Pxr1 and Tma23 predicted G-patch position is displayed in red. The x-ray structure of DHX15:NKRF$^G$ (PDB 6SH6)[25] is shown for comparison and the winged-helix domain (WH) is marked. **b** Difference in H/D exchange rate between Prp43 alone or when in complex with either Pxr1$^{GM}$ (red) or Tma23$^{GM}$ (blue). Each point represents a peptide, plotted according to its centroid value. The Y axis shows the H/D difference calculated as the product of the difference in percentage deuteration multiplied by the difference calculated in number of deuterons (%D * #D). Peptides shown in the uptake plots (**c**) are indicated above the plot (the length of the lines is not in scale).

**c** Uptake plots for Prp43 peptides representing the different regions (indicated above each plot) that show significant differences in H/D exchange rates in different conditions. Peptides 296-307, 505-516, and 627-634 represent statistically non-significant (n.s.) H/D exchange rates. H/D exchange rates for Prp43, Prp43-Pxr1$^{GM}$, and Prp43-Tma23$^{GM}$ are shown in black, red, and blue, respectively. Stars mark the conditions for which H/D exchange of Prp43 was statistically significant ($p < 0.05$) in presence of Pxr1$^{GM}$ (red) or Tma23$^{GM}$ (blue). A two-sided t-test student was applied for data analysis; three technical replicates are presented with mean values ± SD (See also Supplementary Data. 1). **d** Crosslinks of Prp43-Tma23$^{GM}$ complex were mapped on the AlphaFold2 predicted model. The crosslinked amino acids are shown in sticks. The cross-linked pairs are indicated in the same colour, connected with a dashed lined, and summarized in the table. The I-patch is shown in mesh representation. See also Supplementary Data. 2.

replaced Pxr1$^D$ at the genomic locus with CoV2$^C$ within the Y2H reporter strain. In this modified strain (Fig. 6a, right panel, Pxr1$^{GM}$-CoV2$^C$), we find that full-length Pxr1$^{F126A}$ interacts with Ppr43 (Fig. 6a, right panel, row 10). Attempts to tag endogenous Tma23 with CoV2$^C$ in the Y2H reporter strain were unsuccessful. Hence, we tested interactions between Tma23$^{GM}$-Pxr1$^D$/Tma23$^{GM-F96A}$-Pxr1$^D$ chimeric fusions with Prp43 in the Pxr1$^{GM}$-CoV2$^C$ strain (Fig. 6a, right panel). We find that both constructs interact with Prp43 (Fig. 6a, right panel, rows 11 and 12). The

interaction of Tma23$^{GM-F96A}$-Pxr1$^D$ fusion with Prp43 is modestly impaired. Notably, the Pxr1$^{F126A}$ and Tma23$^{F96A}$ mutants were growth impaired, but did not exhibit a severe defect like Pxr1- and Tma23-depleted cells (Fig. 6c, compare rows 1, 2, and 3; and rows 6, 7, and 8). These data implicate additional interactions between Prp43 and Pxr1/Tma23 besides the I-patch. Curiously, strains expressing Pxr1$^{GM}$-CoV2$^C$ and Tma23$^{GM}$-CoV2$^C$ chimeras wherein the Phe within the I-patch was substituted to an Ala were strongly growth impaired like Tma23 and

Pxr1-depleted cells (Fig. 6c, compare rows 2, 4 and 5; and rows 7, 9 and 10). This severe growth impairment for the Phe substitution was rescued in the context of the functional Pxr1$^{GM}$-Tma23$^D$ and Tma23$^{GM}$-Pxr1$^D$ chimeras (Fig. 6d, rows 6 and 12). These data support the notion that the dimerization segments (Tma23$^D$ and Pxr1$^D$) enable additional contacts with Prp43, which are lost in the context of the CoV2$^C$-driven homodimerization.

We employed the 60S reporter, uL18-GFP, to quantitatively evaluate the phenotypes underlying the synergistic interactions described above. While the Pxr1$^{F126A}$ and Pxr1$^{GM}$-Cov2$^C$ mutants showed a modest nuclear accumulation of uL18-GFP, the Pxr1$^{GM\text{-}F126A}$-Cov2$^C$ mutant exhibits a stronger phenotype (Fig. 6e, upper panel). This synergistic defect is significantly rescued when the heterologous dimerization motif Cov2$^C$ is substituted by Tma23$^D$ (Pxr1$^{GM\text{-}F126A}$-Tma23$^D$; Fig. 6e, upper panel). A similar synergistic defect was observed with the Tma23$^{GM\text{-}F96A}$-Cov2$^C$ mutant, that was significantly rescued in the Tma23$^{GM\text{-}F96A}$-Pxr1$^D$ (Fig. 6e, lower panel).

All the above data are consistent with a role for the I-patch and dimerization module within Tma23 and Pxr1 to enable Prp43 binding during 60S assembly.

## Discussion

An unbiased CRAC approach showed that yeast Prp43 crosslinked to multiple sites within 18S and 25S containing pre-rRNA[22]. How Prp43 is recruited to these sites and how the timing of ATPase activation is regulated locally remains unexplored. The G-patch-containing Pfa1 belongs to a functional network involved in preparing 20S pre-rRNA for endonucleolytic cleavage[21]. Therefore, Prp43 remodelling activity within the 40S pre-ribosome has been suggested to operate via Pfa1-mediated activation[21]. Here, we unveil the nucleolar localized Tma23 as a member of the G-patch family that stimulates Prp43-ATPase activity during 60S assembly.

Amongst the G-patch cofactors co-enriching with Prp43-TAP, we find that Tma23 and another nucleolar localized G-patch cofactor, Pxr1, exhibits a similar domain organization (Fig. 2a, upper panel). Depletion of either Tma23 or Pxr1 in yeast induced severe growth defects and accumulated the 60S reporter, uL18-GFP, in the nucleus (Fig. 3a, b)[16]. Substitution of a conserved leucine within the G-patch brace-helix of Tma23 and Pxr1 (Tma23$^{L7E}$ and Pxr1$^{L33E}$) important to activate Prp43[25] induced growth defects and accumulated uL18-GFP in the nucleus (Fig. 3d, e). These data underscore the requirement for Prp43 activation mediated by Tma23$^G$ and Pxr1$^G$ during 60S assembly. Pxr1 and Tma23 are unable to substitute each other's function in vivo (Fig. 6d) indicating that these G-patch-containing proteins activate Prp43 at different yet unknown locations on the 60S pre-ribosome.

We find that the three G-patches (Pfa1$^G$, Pxr1$^G$, and Tma23$^G$) that function during ribosome assembly stimulate Prp43-ATPase activity to different extents. Biochemical studies show that Pfa1$^G$ forms a stable complex with Prp43, in contrast to Tma23$^G$ and Pxr1$^G$. This altered affinity seems to underlie the strong stimulation of Prp43-ATPase activity by Pfa1$^G$, as compared to Tma23$^G$ and Pxr1$^G$. Consistent with this, fusing Tma23$^G$ directly to Prp43 (Prp43-T$^G$) resulted in a hyper-stimulation of its ATPase activity (Fig. 2e). Amongst the three G-patches, Pxr1$^G$ seems to be the weakest activator as the Prp43-P$^G$ fusion showed a modest stimulation of ATPase activity (Fig. 2f). The differences explain the limited portability of the G-patches between the different Prp43 cofactors involved in pre-mRNA splicing (Spp382) and ribosome assembly (Pfa1, Pxr1)[23]. The differential activation by the G-patches might reflect the complexity of the environment of the target RNA substrate that needs to be remodelled by Prp43.

Y2H and biochemical studies showed that Tma23$^M$ and Pxr1$^M$ adjoining the G-patch forms stable complexes with Prp43. Initially, we suspected that Tma23$^M$ and Pxr1$^M$ may be responsible to bring Prp43 in the vicinity of the G-patch to locally stimulate its ATPase activity. Instead, we find that Tma23$^M$ and Pxr1$^M$ restrains Prp43-ATPase activity. We find that this inhibitory activity lies within a 14-residue segment in Pxr1$^M$ (Fig. 4d), which we term I-patch. Cryo-EM and HDX-MS studies show that Prp43 interacts with the I-patch using an interaction platform that does not clash with the G-patch binding site(s) (Fig. 5a, b). These data indicate that the I-patch does not compete with the G-patch to inhibit Prp43 ATPase-activity. How does the I-patch then restrain Prp43-ATPase activity? Given that the active site of Prp43 is located between RecA1 and RecA2 domains, alteration in the motion of the RecA-like domains should impact the geometry of the nucleotide binding site and consequently ATPase-activity[11]. I-patch interacts with the β-strand on the surface of RecA2 which continues as a loop connecting the RecA-like domains, and later as a β-strand within the RecA1 domain (Fig. 5b). This structural element within RNA helicases renders the RecA1 and RecA2 domains flexible relative to each other. The G-patch has been proposed to stimulate Prp43-ATPase activity by restricting the mobility of the RecA-like domains in a productive manner[25] and also replenishing the active site with ATP by promoting ADP release[26]. We find that I-patch strongly reduces the $k_{cat}$ values associated with Prp43-ATPase activity. Given that the $K_M$'s for Prp43, Prp43:Tma23$^M$ and Prp43:Pxr1$^M$ are in a similar μM range, we infer that the affinity of ATP toward Prp43 might not be affected. Instead, we speculate that I-patch binding may alter the rigidity of the β-strand-loop-β-strand to trap the two RecA-like domains of Prp43 in an unproductive conformation that disfavours ATP hydrolysis.

What could be the function of the I-patch during ribosome assembly? Given the complexity of ensuring correct pre-rRNA folding especially during early nucleolar biogenesis steps, it is conceivable that the timing of local Prp43 activation is tightly controlled to prevent premature processive remodelling. Prp43 could be recruited to the pre-ribosome, where it is held in an inactive state by I-patch. Only when the RNA substrate is ready for remodelling, the inhibitory hold of the I-patch on Prp43 is released by a yet unknown mechanism to permit activation by the G-patch. I-patch may serve as a brake to coordinate Prp43 activity with other RNA remodelling steps occurring concurrently on the assembling pre-ribosome. We propose that Prp43-ATPase activity is tuned locally through toggling interactions with the G-patch and the I-patch.

We find that the C-terminal segments of Tma23 and Pxr1 (Tma23$^D$ and Pxr1$^D$) are functionally equivalent and homodimerize. These segments can be exchanged between Tma23 and Pxr1 and can even be replaced by a heterologous CoV2$^C$ dimer (Fig. 6c, d). Curiously, CRAC studies show that the different crosslinked sites of Prp43 are positioned as pairs within the 25S regions of the pre-rRNA[22,28]. These data support the notion that dimeric Prp43 complexes operate during 60S assembly. However, which pairs are targeted by Tma23 and Pxr1 remains to be determined. Y2H, biochemical and genetic interactions support a role for the dimerization module to enable interactions between Prp43 and the I-patch. How the dimerization module enables Prp43 binding remains unclear.

Given that the I-patch docks on a ubiquitous β-strand-loop-β-strand that connects the RecA-like domains, it is likely that other DEAH RNA helicases may employ this platform to regulate their ATPase-activities. Interestingly, the Ski2-like RNA helicase Mtr4, that is a key component of the RNA degradation machinery, interacts with its cofactor NRDE2 using the same β-strand (PDB 6IEH)[52]. Moreover, AlphaFold2-based modelling studies predict I-patch segments within the metazoan GPATCH3[12] and GPATCH4[12,53] that interact with the same structural element within DHX15 (Supplementary Fig. 1f). We anticipate that AI-guided structural modelling combined with experimental validation will expand the inventory of I-patches that tune the ATPase-activity of RNA helicases involved in different aspects of gene expression.

## Methods

### Yeast strains and plasmids

All *Saccharomyces cerevisiae* strains used in this study are listed in Supplementary Table 2. Genomic disruptions, promoter switches, and C-terminal tagging at genomic *loci* were performed using standard yeast molecular biology[54]. All plasmids used in this study are listed in Supplementary Table 3. All recombinant DNA work was performed using *Escherichia coli* TOP10 cells. Gene mutations were generated with QuickChange site-directed mutagenesis kit (Agilent Technologies, Santa Clara, CA, USA). Prp43-T[G], Prp43-P[G], Tma23[GM]-CoV2[C] and Pxr1[GM]-CoV2[C] fusions were synthesized by Twist Bioscience (San Francisco, CA). All cloned DNA fragments and mutagenized plasmids were verified by Sanger-sequencing.

### Fluorescence microscopy

To assess ribosome maturation defects, the $P_{GAL1}$-*TMA23* and $P_{GAL1}$-*PXR1* strains were transformed with uS5-eGFP and uL18-GFP. The pre-cultures were grown to saturation in YPG. Cells were diluted to $OD_{600} = 0.2$, shifted to YPD, and let grown until mid-log phase. Cells were harvested and the pellet was washed once in $dH_2O$. 3 μl of cells were transferred on a microscopy slide (VWR) and covered with a glass slip (18×18 mm no 1, VWR). Cells were visualized using Leica DMi8 Thunder Imager 3D microscope (Leica, Germany) equipped with HC PL APO 63x/1.40-0.60 oil immersion objective (Leica, Germany). Images were acquired with a fitted digital camera Leica DFC9000 GT (Leica, Great Britain) and processed with ImageJ software (version 1.50e; NIH and LOCI, USA).

### Tandem-Affinity purification

Two litres of TAP-tagged yeast strain cultures were grown in YPD to $OD_{600}$ 3-3.5 and harvested at 7100 x$g$, for 15 min at 4 °C. The cells were transferred to a 50 ml tube, washed with 50 ml $dH_2O$, pelleted at 7100 x$g$, for 5 min at 4 °C, and resuspended in lysis buffer (50 mM Tris-HCl pH 7.5, 1.5 mM $MgCl_2$, 0.15% (v/v) Igepal CA-630, 75 mM NaCl), complemented with ½ tablet of complete EDTA-free protease inhibitor cocktail (Roche), 1.5 mM PMSF and 1 mM DTT in a final volume of 25 ml. Cells were lysed at 500 rpm for 20 min at 4 °C with glass beads (400-600 mm diameter) in a Pulverisette 6 planetary mill (Fritsch, Germany). The lysate was separated from the glass beads by applying the lysate to a 50 ml syringe and pushed down into a new 50 ml tube. The lysate was clarified by centrifugation at 4500 $g$ for 10 min at 4 °C, followed by centrifugation at 38,800 x$g$ for 30 min at 4 °C. The clear cell lysate was recovered into a new tube and 150 μl of equilibrated IgG Sepharose beads (Cytiva, Sweden AB) were added and incubated for 1.5 h at 4 °C, on a rotating wheel. The beads were applied to a 10 ml column (Bio-Rad) and washed twice with 5 ml lysis buffer with 0.5 mM DTT. Elution was made by incubating the beads with TEV protease in 1 ml lysis buffer complemented with 0.5 mM DTT, overnight at 4 °C on a rotating wheel. The TEV eluate was collected into a new 10 ml column. The IgG Sepharose beads were washed with 5 ml lysis buffer (no DTT) and the flow-through was collected into the second 10 ml column. 300 μl of equilibrated Calmodulin Sepharose beads (Cytiva, Sweden AB), 1 mM $CaCl_2$ and 1 mM DTT were added to the column and incubated at 4 °C for 1 h on a rotating wheel. The beads were washed with 10 ml lysis buffer containing 1 mM $CaCl_2$ and 1 mM DTT. Bound proteins were eluted by incubation at 35 °C four times for 10 min in 300 μl of elution buffer (10 mM Tris-HCl pH 7.5, 50 mM NaCl, 5 mM EGTA, final volume: 1.2 ml). The proteins were precipitated by adding 10% (v/v) trichloroacetic acid (TCA) and incubated on ice for 15 min. Precipitated proteins were centrifuged at 18,400 x $g$ at 4 °C for 10 min, washed with 1 ml ice-cold acetone and centrifuged again under the same conditions. The pellet was air-dried and resuspended in 50 μl 1x LDS sample buffer (Invitrogen). Samples were heated for 10 min at 70 °C and separated on NuPAGE 4-12% Bis-Tris gradient gels (Invitrogen) and applied to silver staining, Western blotting or sent for label-

free quantitative mass spectrometry. Affinity-purified proteins were heated for 10 min to 70 °C and centrifuged at 18 400x$g$. 1 – 10 μl of the supernatant was loaded on 15% SDS-polyacrylamide gels, SurePAGE Bis-Tris 4-20% gradient gels (Genscript, EUA) or onto NuPAGE Novex 4-12% gradient gels (Invitrogen). For SDS-polyacrylamide gels SurePAGE Bis-Tris 4-20%, proteins were separated at 120 V for 1 h 20 min and subjected to Western blotting. Primary polyclonal antibodies against Tma23 and Pxr1 were raised in rabbits using the following peptide sequences: DGEAWWERLFDGHLKNLDV and KQKRAALMDSKAL-NEIFM, respectively. Affinity-purified anitbodies were used in a 1:1000 dilution for Western blotting. Primary polyclonal antibodies against recombinant Prp43 were raised in rabbits, and the serum with a 1:500 dilution for Western blotting. α -CBP (Merck, AG; catalogue number #07-482) was used in a dilution of 1:1000. Secondary antibody (HRP-conjugated α-rabbit; Merck AG; catalogue number #A0545) was used at 1:1000 dilution. For NuPAGE Novex 4-12% gradient gels (Invitrogen), proteins were separated at 150 V for 1 h 15 min in MOPS buffer (50 mM MOPS pH 7.7, 50 mM Tris-base, 0.1% SDS, 1 mM EDTA) and subjected to silver staining.

### Mass Spectrometry of TAP-purified proteins

The acquired MS raw data were processed for identification and quantification using FragPipe (version 16.0), MSFragger (version 3.3), and Philosopher (version 4.4.0)[55]. The resulting LC-MS data was processed using a scripted Philosopher workflow[56] followed by Label-Free Quantification using IonQuant[57]. In short, MS2 spectra of a minimum of 2 peptides per protein were searched against the Saccharomyces cerevisiae (strain ATCC 204508/S288c) UniProtKB reference proteome (UP000002311, retrieved on 2022-09-05) by the MSFragger search engine 3.4, allowing for one missed tryptic cleavage and fixed carbamidomethylation of Cysteine, variable Methionine oxidation, and variable acetylation of the protein N-terminus after Methionine removal. Label-free quantification was performed applying the FDR-controlled Match-Between-Runs algorithm as implemented by Ion-Quant 1.7.17.

A set of R functions implemented in the R package prolfqua[58] was used to convert the protein razor intensity reported in the combined_protein.txt file into SAINTexpress[59] compatible format. The analysis was run on the local infrastructure compute infrastructure[60]. Finally, the output was filtered for a Bayesian false discovery rate of less than 10% and an empirical fold change score greater than two to produce a list of protein-protein interactors (BFDR < 0.05, EFC > 2). The dataset has been deposited to the ProteomeXchange Consortium via the PRIDE[61] partner repository with the dataset identifier PXD048382.

### Recombinant protein expression and purification

Recombinant proteins were co-expressed in *E. coli* BL21(DE3)* cells at 15 °C overnight, by 0.3 mM IPTG (Panreac Applichem GmbH, USA) (added at ~$OD_{600}$ = 0.5). After harvest, the pellets were resuspended in Hepes buffer (50 mM Hepes pH 7.5, 200 mM NaCl, 5 mM $MgCl_2$, 2 mM DTT), complemented with ½ tablet of cOmplete EDTA-free protease inhibitor cocktail (Roche AG, Basel). Cells were placed on ice and lysed by sonication for 3 min with a Sonic Ruptor 4000 (Omni International, USA). The lysate was clarified by centrifugation at 38 800 x$g$ for 30 min at 4 °C and incubated with preequilibrated Ni-NTA Agarose beads (Qiagen) (previously washed four times) for 1 h at 4 °C on a rotating wheel. $His_6$-tagged proteins were affinity purified by gravity flow in Hepes buffer. The beads were washed with Hepes buffer complemented with 40 mM imidazole and the proteins eluted with Hepes buffer containing 240 mM imidazole. The purified proteins were dialysed in the presence of 0.03 mg/ml RNAse A (Thermo Scientific, MA, EUA) at 4 °C overnight in Hepes buffer (no imidazole) and applied to Size Exclusion Chromatography. The proteins were loaded on a Superose 6 Increase 10/300 GL column (Cytiva) and the appropriate

fractions were collected and subjected to ATPase activity assays, Cryo-Electron Microscopy, Crosslinking Mass-Spectrometry or Hydrogen-Deuterium Exchange Mass-Spectrometry analyses. For ATPase activity assays sample aliquots were flash-frozen and stored at −80 °C in buffer containing 10% (v/v) glycerol.

### Pulldown assays

Recombinantly co-expressed proteins were grown and induced as mentioned previously. Cells were lysed on ice by sonication for 3 min with a Sonic Ruptor 4000 (Omni International, USA) in 25 mM Tris-HCl pH 7.5, 150 mM NaCl, 5 mM MgCl$_2$, 10% glycerol, 0.1% NP-40 (Igepal, Merck, Germany), 10 mM β-mercaptoethanol, 1 mM PMSF, and ½ tablet of cOmplete EDTA-free protease inhibitor cocktail (Roche AG, Basel). The lysate was clarified by centrifugation at 38 800 x $g$ for 30 min at 4 °C. GST-tagged proteins were immobilized on pre-equilibrated Glutathione Sepharose 4 Fast Flow beads (Cytiva) and incubated for 1 h at 4 °C, with rotation. The samples were centrifuged at 380 x $g$ for 6 min at 4 °C and the supernatant was discarded. The pellet was resuspended in 1 ml buffer and transferred to a 1.5 ml tube. The beads were washed 6 times with 1 ml buffer and the bound proteins were eluted with 50 µl 2x LDS, heated up at 70 °C for 10 min. The proteins were separated by SDS-Page on a 15% polyacrylamide gel or SurePAGE Bis-Tris 4-20% gradient gel (Genscript, EUA).

### ATPase activity assays

ATPase activity assays were performed using a protocol adapted from Studer and colleagues[25]. 1 mM phosphoenolpyruvate, 0.8 mM nicotinamide adenine dinucleotide (NADH), 1 mM DTT, 20 mM MgCl$_2$, 12 U of pyruvate kinase, 18 U of lactate dehydrogenase were mixed in the assay buffer: 50 mM Hepes, pH 7.5, 200 mM KCl, 1 mM DTT. 1 µM of each protein or protein complex was used, except when indicated. Reactions were performed in 96-well plates (BRANDplate, pureGrade) in a total volume of 100 µl. The assay was initiated by the addition of 2.5 mM ATP to the mix. Absorption at 340 nm was monitored using Biotek PowerWave XS Microplate Reader (BIO-TEK) for 60 min, at 30 s intervals, at room temperature. The absorbance was used to calculate the NADH concentration, which directly corresponds to the concentration of ATP hydrolysed. A linear regression of the data points was performed. The resulting slope corresponding to the ATP hydrolysed over time was inverted and normalized to the enzyme concentration. The displayed rates correspond to the number of ATP hydrolysed per enzyme per minute. All measurements were performed in triplicates. Background ATPase activities of GST-Tma23$^G$ and GST-Pxr1$^G$ purifications and buffer containing U$_{20}$ RNA were assessed. Those preparations which exhibited initial reaction rates like the buffer alone control were used for the ATPase assays. Kinetic parameters were determined by fitting the Michaelis-Menten equation to the initial rates corresponding to serial 4-fold dilutions of ATP, to a maximum concentration of 2.5 mM. Absorption at 340 nm was monitored using Biotek PowerWave XS Microplate Reader (BIO-TEK) for 60 min, at 49 s intervals, at room temperature.

### K$_D$ determination by fluorescence polarization

Fluorescence anisotropy was measured in non-binding 96-well plates (Greiner Bio-One) using a Tecan Safire II plate reader (Tecan Group Ltd.) equipped with a fluorescence polarization module. All measurements were conducted in 50 mM HEPES, pH 7.5, 50 mM KCl, 5 mM MgCl$_2$, 0.05% Tween-20 and 1 mM DTT. 5 nM Cy5-labelled U$_{12}$-RNA (Microsynth) was titrated with increasing concentrations from 20 pM to 2.5 µM of either Prp43, Prp43-Pxr1$^M$ or Prp43-Tma23$^M$, providing a total of 24 measurement points. The excitation wavelength was set to 635 nm and the emission was measured at 670 nm using a bandwidth of 10 nm. Each titration was performed in four replicates. K$_D$ values were obtained by fitting the normalized averages to a single-site binding model, as described previously[62].

### Yeast two-hybrid assay

Vectors encoding Prp43, and Tma23 and Pxr1 variants fused to a DNA-binding (pLexA Trp marker) and an Activation domain (pAct2.2 Leu marker) were transformed into the NMY32 reporter strain which encodes a HIS3 reporter gene. The interaction between Prp43 and Tma23/Pxr1 (and variants) brings the DNA binding Domain and Activation Domain together, thus enabling the expression of the HIS3-reporter. The interaction is scored (via the expression of the HIS3-reporter) on plates lacking histidine. The strength of the interaction (HIS3-reporter expression levels) is assessed by performing the growth analyses in the presence of increasing concentrations of inhibitor 3-amino-1,2,4-triazole (3-AT). Transformed NMY32 yeast cells were plated in synthetic dextrose medium (SD) lacking leucine and tryptophan (SD-Leu-Trp) and incubated at 30 °C for 3–5 days. The cells were spotted in a 10-fold serial dilution on SD-Leu-Trp, SD-His, and SD-His supplemented with 3-AT (Merck) at indicated concentrations and incubated at 30 °C for 2–5 days.

### Chemical crosslinking mass spectrometry

Prp43:Tma23$^{GM}$ and Prp43:Pxr1$^{GM}$ complexes were crosslinked at 20 µg scale in 50 mM HEPES pH 7.5, 200 mM NaCl, 5 mM MgCl$_2$, 2 mM DTT buffer. Total protein concentrations were approximately 0.3 mg/ml for Prp43:Pxr1$^{GM}$ and 0.24 mg/ml for Prp43:Tma23$^{GM}$. Crosslinking was performed with 0.25 mM or 1 mM light/heavy disuccinimidyl suberate (DSS-d0/d12, Creative Molecules)[63,64]. Crosslinking was performed for 30 min at 37 °C before quenching the reaction by addition of ammonium bicarbonate to 50 mM final concentration.

### Sample preparation for mass spectrometry

Quenched samples were dried in a vacuum centrifuge and redissolved in 8 M urea, followed by cysteine reduction and alkylation steps (2.5 mM tris-(2-carboxyethyl) phosphine for 30 min at 37 °C and 5 mM iodoacetamide for 30 min at room temperature in the dark, respectively). The samples were diluted to approximately 5.5 M urea with 150 mM ammonium bicarbonate and 200 ng endoproteinase Lys-C (Fujifilm/Wako) was added. After incubation for 2.5 h at 37 °C, samples were further diluted to 1 M urea with 50 mM ammonium bicarbonate, and 400 ng trypsin (Promega) was added. Samples were further incubated at 37 °C and acidified by adding concentrated formic acid to 2% final concentration after overnight digestion. Desalting was performed using Sep-Pak tC18 cartridges (Waters).

### Liquid chromatography-tandem mass spectrometry (LC-MS/MS)

Purified samples were analyzed by LC-MS/MS on a system consisting of an Easy nLC-1200 HPLC and an Orbitrap Fusion Lumos mass spectrometer equipped with a Nanoflex electrospray source (all Thermo Fisher Scientific). Approximately 1 µg of peptides were separated on a PepMap RSLC C18 column (250 mm × 75 µm, Thermo Fisher Scientific) at a flow rate of 300 nl/min. Mobile phases were A = water/acetonitrile/formic acid (98:2:0.15, v/v/v) and B = acetonitrile/water/formic acid (80:20:0.15, v/v/v), and the gradient was set from 11 to 40% B in 60 min.

MS/MS data were acquired in data-dependent acquisition/top speed mode using a cycle time of 3 s. MS1 spectra were acquired at a nominal resolution of 120000 at m/z 200, and precursors with a charge state between 3+ and 7+ were selected for collision-induced dissociation in the linear ion trap with 35% normalized collision energy. Fragment ions were detected in the linear ion trap with the rapid speed setting.

### Data analysis

Data in Thermo Fisher raw format were converted to the mzXML format using msconvert, part of ProteoWizard[65]. Crosslinked peptides were identified using xQuest, version 2.1.5 (available from https://gitlab.ethz.ch/leitner_lab/xquest_xprophet)[63,66,67]. The databases

consisted of the sequences of the respective target proteins and the UniProt-derived sequence of *ARNA_ECOLI* that was identified as a contaminant. A shuffled version of the database was used as a decoy database. Search parameters included: enzyme = trypsin with up to two missed cleavages except the crosslinking site, precursor mass error = 15 ppm, fragment mass error = 0.2 Da for common ions (not containing the crosslinking site) and 0.3 Da for xlink ions (containing the crosslinking site). Crosslink reaction specificity was set to Lys and protein N terminus. Methionine oxidation was specified as a variable modification, while carbamidomethylation of cysteine was set as a fixed modification.

Primary search results were further filtered with narrower MS1 mass error tolerance (−5 to 0 ppm), a minimum TIC value of 0.1, a delta score of ≤0.9 and a mions score of ≥3. Using an xQuest score cut-off of 16, the estimated false discovery rate (FDR) was less than 5% for intra-protein links at the level of non-redundant peptide pairs. A reliable estimate of inter-protein FDR was not possible due to the small number of hits. All inter-protein candidate crosslinks were manually evaluated and are summarized in Supplementary Data 2. The purified complexes were crosslinked in two independent experiments using two different reagent concentrations, and each sample was analyzed by mass spectrometry. The dataset has been deposited to the ProteomeXchange Consortium via the PRIDE[61] partner repository with the dataset identifier PXD047173.

## Hydrogen-Deuterium Exchange mass spectrometry

HDX-MS experiments were performed at the UniGe Protein Platform (University of Geneva, Switzerland) following a well-established protocol with minimal modifications[68]. Details of reaction conditions and all data are presented in Supplementary Data 1. HDX reactions were done in 50 µl volumes with a final protein or protein-complex concentration of 1.8 µM. Briefly, 90 picomoles of Prp43 alone or in complex were diluted to 10 µl in aqueous buffer (20 mM Tris pH 7.5, 200 mM NaCl, 5 mM MgCl$_2$, 2 mM DTT) and pre-incubated on ice for 5 min before initiation of deuteration reactions.

Deuterium exchange reaction was initiated by adding 40 µl of D$_2$O exchange buffer (10 mM Tris pH 7.5, 200 mM NaCl, 5 mM MgCl$_2$, 2.5 mM ATPγS in D$_2$O) to the protein sample. Reactions were carried-out on ice for 3 incubation times (3 s, 30 s, 300 s) and terminated by the sequential addition of 20 µl of ice-cold quench buffer 1 (4 M Gdn-HCl, 1 M NaCl, 100 mM NaH$_2$PO$_4$ pH 2.4, 1% formic acid [FA]). Samples were immediately frozen in liquid nitrogen and stored at −80 °C for up to two weeks. All experiments were repeated in triplicate.

To quantify deuterium uptake into the protein, samples were thawed and injected in a UPLC system immersed in ice with 0.1% FA as the liquid phase. The protein was digested via two immobilized pepsin columns (Thermo Fisher Scientific), and peptides were collected onto a VanGuard precolumn trap (Waters). The trap was subsequently eluted, and peptides separated with a C18, 300 Å, 1.7 µm particle size Fortis Bio 100 × 2.1 mm column over a gradient of 8–30 % buffer C over 20 min at 150 µl/min (Buffer B: 0.1% formic acid; buffer C: 100% acetonitrile). Mass spectra were acquired on an Orbitrap Velos Pro (Thermo Fisher Scientific), for ions from 400 to 2200 m/z using an electrospray ionization source operated at 300 °C, 5 kV of ion spray voltage. Peptides were identified by data-dependent acquisition of a non-deuterated sample after MS/MS and data were analyzed by Mascot. All peptides analysed are shown in Supplementary Data 1. Deuterium incorporation levels were quantified using HD examiner software (Sierra Analytics), and the quality of every peptide was checked manually. Results are presented as percentage of maximal deuteration compared to theoretical maximal deuteration. Changes in deuteration level between the two states were considered significant if >9 % and >0.7 Da and $p < 0.05$ (unpaired t-test). The dataset has been deposited to the ProteomeXchange Consortium via the PRIDE[61] partner repository with the dataset identifier PXD046892.

## Single particle cryo-electron microscopy

10 µl of 1 mg/ml purified Prp43-Pxr1$^{GM}$ complex was diluted by addition of 37.5 µl buffer (50 mM Hepes pH 7.5, 200 mM NaCl, 5 mM MgCl$_2$, 2 mM DTT). Subsequently, 2.5 µl of 100 mM ATPγS stock solution was added, bringing the protein to a final concentration of 0.2 mg/ml with 5 mM ATPγS. The protein complex was vitrified immediately after mixing. The climate chamber of a Thermo Fischer Scientific Vitrobot was equilibrated to a temperature of 4 °C and 95% humidity. A volume of 4 µl sample was applied to a Quantifoil R1.2/1.3 holey carbon grid and subjected to a vitrification protocol using 10 s pre-blot incubation followed by 2 s or 5 s blotting, and subsequent rapid transfer into liquid ethane-propane mix. Micrographs were collected on a Thermo Fisher Scientific Titan Krios cryo-electron microscope equipped with a Gatan K3 direct electron detector and a GIF BioContinuum energy filter operated with a slit width of 20 eV. Images were acquired in counted super-resolution mode and fractionated into 40 frames. Two datasets were acquired: for dataset 1, 8686 movies were collected at a magnification of 105,000x (pixel size: 0.84 Å/pixel) with an electron dose of 85 e⁻/Å$^2$. For dataset 2, 15,047 movies were collected at a magnification of 130,000x (pixel size: 0.66 Å/pixel) with an electron dose of 80 e⁻/Å$^2$. All image processing was carried out in cryoSPARC (version v3.3.1)[69]. The image processing workflow is shown in Supplementary Table 1 and Supplementary Fig. 3. Micrographs were gain corrected, drift corrected, dose weighted and averaged using Patch Motion Correction. Defocus was subsequently determined by using Patch CTF Estimation. 4 732 (dataset 1) and 13 708 (dataset 2) movies were selected. Manually picked particles were used to train a neural network for particle picking in Topaz[70], which was further improved by training with selected particles after 2D classification. Particle picking with Topaz resulted in 822,396 particles (dataset 1 with 0.84 Å/pixel) and 2,265,329 particles (dataset 2; 0.66 Å/pixel), with a combined number of 3,087,725 particles. Particles were extracted with box sizes of 264 pixels (dataset 1) and 336 (dataset 2) and scaled to the same box size of 154 pixels (pixel size: 1.44 Å/pixel) to combine the two datasets. Particles were subjected to four 2D classifications into 200 classes. The purpose of the first 2D classification was the removal of high-contrast junk and empty particles; during the first 2D classification, 1,642,708 particles (53%) were retained. A second 2D classification was performed and 611 291 particles (20% with respect to the initial particle count) were retained. After the second 2D classification, classes that display clearly recognizable protein shapes (including Prp43 and GST) in the class average were selected and subjected to a third 2D classification. In the third and fourth 2D classification, classes whose class average clearly showed Prp43 were selected, including only classes whose class averages exhibited secondary structure features after the fourth classification. 536,414 particles were retained after the third 2D classification (17% with respect to the initial particle count), 444,485 particles were retained after the fourth classification (14% with respect to the initial particle count). An initial model was generated from the particles selected after the fourth 2D classification, using Ab-initio Model Generation (one class). Additional initial models (3 classes) were generated from particles that were not selected after the third 2D classification. These four initial models were used to classify particles selected after the third 2D classification into four classes using Heterogeneous Refinement. One class containing 259,523 particles yielded a reconstruction to 3.39 Å resolution and was used for further processing. Particles were re-extracted and brought to the same pixel size: particles from dataset 1, collected at a pixel size of 0.84 Å/pixel, were extracted with a box size of 264 pixels without binning while particles from dataset 2, collected at a pixel size of 0.66 Å/pixel, were extracted with a box size of 336 and binned to 264 pixels. The combined particle sets were refined using Homogeneous refinement and yielded a map with 3.51 Å resolution, which was improved by non-uniform refinement to 3.43 Å resolution. A tight, soft mask around Prp43-Pxr1 density was generated and used for a final cycle of Local Refinement (pose/shift

gaussian prior, 5 extra final passes, maximum align resolution of 2 Å). The resulting map showed further improvement to 3.33 Å resolution.

## Model building and refinement

An atomic model of *S. cerevisiae* Prp43 was generated with AlphaFold[39] and rigid-body docked into the Prp43-Pxr1 cryo-EM density map with Chimera[71]. The position of the C-terminal domains with respect to the RecA domains was adjusted by rigid body fitting in Chimera. The linker between the C-terminal domains and the RecA domains was rebuilt in coot. Pxr1 was built manually in coot. The atomic model of the Prp43-Pxr1 complex was refined with Phenix[72] real-space refine, using morphing, global minimization, local grid search, adp, nqh-flips and occupancy refinement. The structure of *S. cerevisiae* Prp43 with ADP (PDB 2XAU)[7] was used as a reference model during refinement.

## Statistics and reproducibility

Unless otherwise indicated, error bars represent the mean ± standard deviation of at least three biological replicates (i.e., $n \geq 3$). Statistical analysis was performed using Prism (version 10.3.0; GraphPad Software Inc., La Jolla, CA, USA) or Microsoft Excel (Microsoft Office, version 16; Microsoft Corporation, Redmond, WA, USA). Experiments in Figs. 1a and c; 2b–f; 3b and e; 4a–h; 6b and e; and Supplementary Fig. 2b were performed at least three times and one representative result was used for figure preparation.

## Reporting summary

Further information on research design is available in the Nature Portfolio Reporting Summary linked to this article.

## Data availability

The mass spectrometry dataset of Prp43-TAP generated in this study has been deposited in the ProteomeXchange Consortium via the PRIDE[61] partner repository under accession code PXD048382. The XL-MS data are provided in the Supplementary Data 2 file and available in the PRIDE[61] repository under accession code PXD047173. The HDX-MS data generated in this study are provided in the Supplementary Data 1 file and deposited in the ProteomeXchange Consortium via the PRIDE[61] partner repository, under accession code PXD046892. The Cryo-EM density map was deposited in the EMDB database, under accession code EMD-19078. The refined atomic model was deposited in PDB and is accessible with the accession code 8RDY. Maps and oligonucleotide-based cloning/mutagenesis strategies for all plasmids used in this study are available upon request. All Source data is provided with this paper. Source data are provided with this paper.

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

## Acknowledgements

We thank M. Peter for generously sharing yeast strains. We thank D. Kressler for sharing unpublished results; K. Weis, R. Pillai, and the Panse laboratory for discussions, the ZMB, and UZH for maintaining the imaging equipment, and FGCZ for proteomic analysis. V. Panse is supported by the Swiss National Science Foundation [188527], NCCR RNA & Disease [182880], and a Starting Grant Award from the European Research Council [260676 EURIBIO]. We thank the proteomics platform from the Faculty of Medicine at the University of Geneva for HDX-MS data acquisition. A. Leitner thanks P. Picotti for access to instrumentation and laboratory infrastructure, and financial support by the NCCR RNA & Disease [51NF40-182880]. We thank the Scientific Centre for Optical and Electron Microscopy (ScopeM) for access to microscopes at ETH Zurich, and M. Peterek (ScopeM) for technical support.

## Author contributions

Experimental design: D.P.-C., A.G., J.R., O.V., P.K-N., A.L., D.B., V.G.P.; experimental execution: D.P.-C, A.G., J.R., O.V., J.M., M.O.-O., F.R., E.M., A.L.; data analysis: D.P.-C, A.G., J.R., O.V., J.M., M.O.-O., E.M., A.L.; scientific input: O.V., A.L., D.B., V.G.P.; supervision: V.G.P.; writing: D.P.-C., J.R., O.V., A.L., V.G.P.

## Competing interests

The authors declare no competing interests.
