## [Transparent Peer Review file · Nature Communications]

An inhibitory segment within G-patch activators tunes Prp43-ATPase activity during ribosome assembly

Corresponding Author: Professor Vikram Panse

Version 0:

Reviewer comments:

Reviewer #1

(Remarks to the Author)

The authors have searched for Prp43-interacting partners and identified G-patch-containing Tma23 as a Prp43 co-factor. They provide evidence that Tma23 stimulates the ATPase activity of Prp43 and that the integrity of the G-patch is critical for activation. They show that Tma23 depletion or mutation of its G-patch prevents the assembly of export-competent 60S subunit precursors. Most interestingly, they identify an inhibitory segment within both Tma23 and Pxr1, another G-patch activator of Prp43, that inhibits Prp43 ATPase activity. They term this segment the "I patch". Using cryo-EM and Hydrogen-Deuterium Exchange coupled to Mass Spectrometry (HDX-MS) and Crosslinking Mass Spectrometry (XL-MS), they establish how Pxr1 I patch interacts with the RecA2 and RecA1 domains of Prp43 through a set of hydrophobic contacts. Finally, they provide evidence that both Tma23 and Pxr1 can homodimerize. This study is carefully performed and employs state of the art experimental approaches. It reveals a new functional aspect of Prp43 G-patch co-factors and underscores how complex Prp43 regulation can be. Overall, it will be of great interest to scientists working on ribosome biogenesis, RNA helicases and RNA processing in general.

My only reservation is that I feel the authors should have explored the role of the I patch, and possibly that of the dimerization domain, in ribosome biogenesis, since they propose that the I patch plays a crucial role in timely Prp43 regulation. They have at their disposal point mutants in the I patch that inhibit its interaction with Prp43 and that elicit a growth defect. They could for example use cells expressing Pxr1(F126A) or Tma23(F96A) to investigate the role of the I patch in ribosome synthesis: export assays, pre-rRNA processing analyses etc...and compare the effects to those obtained with a depletion of Pxr1 or Tma23.

Minor points:

- Page 7: (Fig. 4b, compare columns 2 and 5): should be Fig. 4a
- Page 13: (Fig. 6c, rows 4 and 11): should be Fig. 6d, rows 6 and 12.
- Legend is missing for Supplementary Fig. 1f.

Reviewer #2

(Remarks to the Author)

Summary

The authors of the paper focus on an ATP-dependent RNA helicase, Prp43, and the activators that tune its function to regulate the processive remodeling of RNA:protein complexes through activating G-patch segments and inhibitory I-patch segments. Specifically, the authors identified the novel G-patch activator Tma23 and compared it alongside the characterized Pxr1 activator to evaluate their effects on Prp43 structure and ATPase activity. By utilizing in-vivo and in-vitro pulldown assays followed by enzyme-coupled assays, the researchers identified the key segments of both Tma23 and Pxr1 that interact with Prp43 to modulate ATPase activity. Through cryo-EM studies and mutational assays, the authors characterized the structural interactions between Prp43 and both activators and concluded that the activating and inhibitory patches exhibited distinct binding patterns with Prp43. This data was further validated through HDX-MS and XL-MS analyses, which highlighted specific interactions across the RecA1 and RecA2 domains of Prp43 that corroborated the structural interactions derived from the cryo-EM analysis and alluded to potential allosteric effects. Lastly, using in-vivo

mutational and growth analyses, the researchers evaluated the role of homodimerization of both Tma23 and Pxr1 in their interactions with Prp43. The authors' work is notable in that they identified a novel Prp43 G-patch activator, Tma23, and specifically characterized the binding interfaces between Tma23/Pxr1 and Prp43. Thus, the authors concluded the presence of allosteric inhibition that modulated the binding of Prp43 to its RNA substrate and its subsequent enzymatic activity. The paper is overall very well written, and the authors' conclusions are generally supported by their empirical results and the biochemical context of the Prp43 system provided throughout the paper.

Recommendations for revisions

Content

Page 10, "Interaction of Prp43 with Pxr1GM and Tma23GM revealed by HDX-MS analyses": I'm curious why HDX-MS data was only shown for Prp43 and not for peptides belonging to Tma23 or Pxr1? Could HDX results validate the 43- or 13-residue segments the authors identified previously as I-patches for each activator? Could this allow the G-patches of each activator to be more precisely identified despite the lack of cryo-EM data?

Page 12, "The C-terminal segments organize Pxr1 and Tma23 homodimers": This section reads as out of place with respect to the remainder of the results section. Compared to their in-vivo, cryo-EM, and HDX-MS studies, this analysis on the homodimerization of Tma23 and Pxr1 does not seem as novel or significant. Besides identifying the dimerization interface of each protein, I believe further research into the structural details of this interaction, possibly using HDX-MS, cryo-EM, and other techniques used previously, would convey more importance to their dimerization analysis. Alternatively, the authors could expand on the biological significance of dimerization and further emphasize future directions of research.

Figures

Figure 2a: There needs to be more details provided to explain the system the authors are using for the yeast two-hybrid assays. Consider adding a diagram for the pLexA and pAct2.2 vectors or a more detailed methods section on this system to further support the results. This inclusion would be helpful for understanding the use of both SD-Leu-Trp and SD-His plates, as well as offering more support for the observed interactions between Prp43 and Tma23/Pxr1.

Figures 2c-2f: Please include an explanation of the y-axis of the ATP hydrolysis figures, similar to the explanation given in the methods section, and how they relate to the reduction in A340 as detailed in the text.

Figure 3a: Please explain the importance of temperature as a variable in the growth assays.

Figure 4b: Is the change in ATPase activity between Prp43-TG + TM43 and Prp43-TG + TM statistically significant? How can the 43-residue region, which the authors claim is the key inhibitory region of Tma23, show comparable if not slightly elevated levels of ATPase activity compared to Prp43-TG + TM?

Figure 4d: The authors' identification of the minimal inhibitory region of Pxr1M13 could be better supported by comparing it to the reduction in ATPase activity of Prp43-TG + PM, similar to that shown in figure 4b for Prp43-TG + TM43 and Prp43-TG + TM.

Figure 7b: Please specify either in the figure or the legend that the uptake curves are for peptides belonging to Prp43 and not the interacting activators. Perhaps move supplementary figure 5c to this figure? Additionally, consider specifying further in which regions (shown in 7a) each set of peptides belongs to. It's never explained which structure corresponds to cs.1 and cs.2, making it very confusing to connect the uptake curves to the structures provided.

Supplementary figure 1d, 1e: Please add pLDDT scores for the provided AlphaFold models to support the use of these models later in the paper. I'm skeptical of AlphaFold's accuracy in modeling the loop regions and so it needs to be quantified.

Supplementary figure 4: There are several instances throughout the paper (pgs. 9, 11) that the structure of Prp43-Tma23G/Pxr1G is compared to DHX15-NKRF, and further conclusions were made based on these similarities. This is the only figure that really addresses these similarities, although I don't believe it offers sufficient evidence to warrant the claims made regarding Tma23G/Pxr1G. Personally, I find this figure and its legend difficult to comprehend due to the variety of RNA helicases mentioned. Perhaps consider editing the figure to more specifically compare the structure of DHX15-NKRF to Prp43-Tma23G/Pxr1G.

Supplementary figure 5c: With the exception of the region 481-491, all of the regions shown in the uptake curves in figure 7 are shown to have relatively low or missing coverage in this figure. This makes me somewhat skeptical of the authors' HDX analysis. Furthermore, it is unclear as to whether this coverage map represents Prp43 bound to Tma23 or Pxr1. The authors state in the legend that it pertains to both bound activators, but having two coverage maps for each activator would be a more effective supplement to the data in figure 7.

(Remarks to the Author)

Portugal-Calisto and colleagues describe a new G-patch-containing factor, Tma23, that plays a role in the modulation of Prp43 activity during ribosome biogenesis. In addition to validating the Tma23 G-patch, their investigation of Tma23 function also uncovered a new structural module, conserved in both Tma23 and another Prp43 modulator, Pxr1, that negatively impacts Prp43 activity. Structural studies of this "I-patch" in complex with Prp43 suggest that it stabilizes an ADP-bound, "open" conformation of Prp43.

Prp43 is involved in a multitude of RNA remodeling functions and understanding how its motor function is specifically targeted and regulated is of broad interest in multiple RNP assembly and catalytic processes. The identification and characterization of new Prp43 regulatory elements is therefore an important finding that merits publication in Nature Communications for a revised manuscript.

There are a few issues that I would like the authors to address:

1. While using non-RNA stimulated ATPase activity is a useful initial assay to define the relative effects of G-patch and I-patch on Prp43 activity, this does not represent the biologically relevant catalytic activity of Prp43, which is translocation along ssRNA. The authors should address why they did not measure the effect of the I-patch (and G-patch) modules on RNA-stimulated ATPase activity. Their high throughput coupled ATPase assay is ideally suited to obtain curves showing the effect of increasing RNA concentration on the activity of Prp43 in the presence of various Tma23 and Pxr1 fragments. These results would not only define the effect of the I-patch in the proper mechanistic context but would also test if the presence of RNA substrate directly promotes I-patch release and subsequent G-patch binding.
2. Along the same lines, why did the authors only measure V_{max} rather than obtaining full kinetic parameters for their Prp43 ATPase assays? Obtaining KM 's of Prp43 ATPase activity with various Tma23 and Pxr1 fragments would allow them to test the hypothesis that the I-patch-engaged state stabilizes the ADP-bound conformation of Prp43.
3. While the use of AlphaFold to model G-patch interactions seems reasonable, the use of AlphaFold to propose a model for Prp43-Tma23GM in Figure 7 seems misguided. The authors show ample evidence that G-patch and I-patch interactions are likely exclusive (i.e. I-patch binds and stabilizes an inactive (ADP), open conformation while the G-patch stimulates the active and closed RNA-bound state). Yet in Figure 7, the AlphaFold model shows the Tma23 I-patch bound to the active, closed state and seems to suggest that G-patch and I-patch interactions can occur simultaneously. My guess is that AlphaFold is biased toward the more compact closed state and cannot predict the conformation stabilized by I-patch binding (The authors could test this by checking if AlphaFold accurately predicts the Prp43/Pxr1M structure, including the RecA1/A2 orientation observed in the cryo-EM structure). More traditional homology modeling is likely a much better strategy to visualize the HDX-MS and XL-MS data, using separate models (and conformations) to model Tma23 G-patch and I-patch interactions with Prp43.
4. Figure 6A: I am concerned that the positive 2-hybrid signal for DBD-Pxr1F126A x Prp43-AD and Tma23F96A x Prp43-AD is being mediated by a "bridging" wild type copy of Pxr1 or Tma23. The only way I can think of controlling for this is to use Pxr1GM-CoV2C and Tma23GM-CoV2C as the "wild-type" genomic copy. The authors need to address this if they want to claim that the D-domain helps with Prp43 binding from 2-hybrid data alone.

Minor issues:

1. Figure 2 – In the ATPase assays, there is no control for background/contaminating ATPase activity for Tma23G, Pxr1G and Pfa1G in the absence of Prp43. I am assuming this background was measured and subtracted, but this is not explicitly stated in the methods. The small Pxr1G stimulatory effect observed in the experiments makes this more important than usual.
2. Figure 3D – Why is this experiment carried out at 37°, when the growth defect associated with Tma23 depletion is much more pronounced at 30° (Fig. 3A)? It would be easier to define if Tma23L7E has an intermediate phenotype at the lower temp.
3. Last paragraph of "An inhibitory patch within...activity"/Figure 4D: The authors should discuss this result more, as it suggests that the effects of the G-patch and the I-patch addition are independent. Was the reciprocal mix and match experiment also performed (i.e. Prp43-PG + Tma23M43)?
4. Cryo-EM sample preparation: Please include a more detailed description of how the sample used for cryo-EM was prepared, especially the relative concentrations of Prp43, Pxr1GM and nucleotide and the order of addition of the sample components.
5. Figure 5: In panel A, please show the actual cryo-EM density (low pass-filtered if needed to show all regions) next to the cartoon model. Panel B – this is a minor point, but the arrangement of these panels (1,2,3) is in the opposite direction of the same regions in the inset of panel A (3,2,1 going right to left). It might be useful to label F126 and L132 in the panel A inset to help with navigation. Also, in panel B2, the presence of ADP is misleading and should be omitted. The perspective makes it look like the nucleotide is quite close when in fact it is probably 15Å away.
- Figure 5C: In the text, the authors state that the "connecting loop moves toward...". I think it would be more accurate to say that the loop adopts a different conformation due to the reorientation of the RecA domains. Even in the figure, it is obvious that the first strand in RecA2 moves as much as the loop.
- Sup. Fig. 4B. It might be confusing to some readers that both nucleotides are labeled ADP, since one is really ADP-BeF3. Please alter in some way to make it clear that one pocket contains ADP and the other "ATP".
6. Supplementary Figure 3: Please add a panel with experimental maps around the Pxr1 peptide as an example of the map quality.

Reviewer comments:

Reviewer #1

(Remarks to the Author)

The authors in my view have introduced all the corrections needed to the text and provided additional data, such that all issues raised by the three reviewers are fully answered. I thus support publication of this extensively revised version.

Reviewer #2

(Remarks to the Author)

The authors of this paper have provided more than sufficient explanations for the questions and points we raised and have thoroughly rectified issues and made appropriate changes to the text and supporting figures. We thank the authors for their detailed responses and receptiveness to our comments, and we support this manuscript to be published.

Reviewer #3

(Remarks to the Author)

The revised version of the manuscript submitted by Portugal-Calisto et al thoroughly addresses my concerns. The new data and analysis included in this version provides a comprehensive characterization of the l-patch, representing an important insight into the spatio-temporal regulation of Prp43. It will be of broad interest to the RNA biology field. I fully endorse publication in Nature Communications.

Reviewer #1 (Remarks to the Author):

The authors have searched for Prp43-interacting partners and identified G-patch-containing Tma23 as a Prp43 co-factor. They provide evidence that Tma23 stimulates the ATPase activity of Prp43 and that the integrity of the G-patch is critical for activation. They show that Tma23 depletion or mutation of its G-patch prevents the assembly of export-competent 60S subunit precursors. Most interestingly, they identify an inhibitory segment within both Tma23 and Pxr1, another G-patch activator of Prp43, that inhibits Prp43 ATPase activity. They term this segment the “I patch”. Using cryo-EM and Hydrogen-Deuterium Exchange coupled to Mass Spectrometry (HDX-MS) and Crosslinking Mass Spectrometry (XL-MS), they establish how Pxr1 I patch interacts with the RecA2 and RecA1 domains of Prp43 through a set of hydrophobic contacts. Finally, they provide evidence that both Tma23 and Pxr1 can homodimerize. **This study is carefully performed and employs state of the art experimental approaches. It reveals a new functional aspect of Prp43 G-patch co-factors and underscores how complex Prp43 regulation can be. Overall, it will be of great interest to scientists working on ribosome biogenesis, RNA helicases and RNA processing in general.** We thank the Reviewer for his/her supportive comments.

My only reservation is that **I feel the authors should have explored the role of the I patch, and possibly that of the dimerization domain, in ribosome biogenesis**, since they propose that the I patch plays a crucial role in timely Prp43 regulation. They have at their disposal point mutants in the I patch that inhibit its interaction with Prp43 and that elicit a growth defect. They could for example **use cells expressing Pxr1(F126A) or Tma23(F96A) to investigate the role of the I patch in ribosome synthesis: export assays, pre-rRNA processing analyses etc...and compare the effects to those obtained with a depletion of Pxr1 or Tma23.**

The discovery of an inhibitory I-patch and a dimerization module within Tma23 and Pxr1 is a highlight of this study. A combination of Y2H, biochemical and genetic studies support a functional interaction between the I-patch and the dimerization module.

Y2H and biochemical studies: Full-length Pxr1^{F126A} interacts with Prp43 (Figure 6a, row 2), however this interaction is lost when the dimerization module is removed (Pxr1^{M-F126A}) (Figure 6a, compare rows 3 & 4; Figure 6b, compare lanes 1 & 2). A similar interaction profile was observed for the Tma23^{F96A} mutant (Figure 6a & 6b).

Genetic studies: The individual Pxr1^{F96A} and Pxr1^{GM-Cov2^C} mutants are not severely growth impaired like the Pxr1-depletion strain (Figure 6c, compare rows 1, 3 & 4). However, when combined, the Pxr1^{GM-F126A-Cov2^C} mutant, exhibits a severe synergistic growth defect (Figure 6c, row 5) like the Pxr1-depletion strain (Figure 6c, row 1). This severe growth defect is partially rescued by substituting the heterologous dimerization motif (Cov2^C) by Tma23^D (Pxr1^{GM-F126A-Tma23^D}) (Figure 6d, row 6). A similar interaction profile was observed for the Tma23^{F96A} mutant (Figure 6c & 6d).

We employed the 60S reporter (uL18-GFP) to evaluate the phenotypes associated with this synergistic interaction. Our genetic interaction studies correlate well with the 60S export assay (Figure 6e). For e.g., while the Pxr1^{GM-F126A} and Pxr1-Cov2^C mutant strains are modestly impaired in 60S export, the Pxr1^{GM-F126A-Cov2^C} mutant shows a severe impairment (Figure 6e, upper panel). This defect is partially rescued in a yeast strain expressing the Pxr1^{GM-F126A-Tma23^D} mutant (Figure 6e, upper panel).

Altogether, these data support the idea that the I-patch and the dimerization module of Tma23 and Pxr1 (Tma23^D & Pxr1^D), together, enable binding to Prp43 in vivo. These data are now shown in Figure 6e and elaborated in the text (page 13, 14).

Minor points:

We thank this Reviewer for pointing out the errors.

- Page 7: (Fig. 4b, compare columns 2 and 5): should be Fig. 4a

Done.

- Page 13: (Fig. 6c, rows 4 and 11): should be Fig. 6d, rows 6 and 12.

Done (now page 14).

- Legend is missing for Supplementary Fig. 1f.

Done.

Reviewer #2 (Remarks to the Author):

Summary

The authors of the paper focus on an ATP-dependent RNA helicase, Prp43, and the activators that tune its function to regulate the processive remodeling of RNA:protein complexes through activating G-patch segments and inhibitory I-patch segments. Specifically, the authors identified the novel G-patch activator Tma23 and compared it alongside the characterized Pxr1 activator to evaluate their effects on Prp43 structure and ATPase activity. By utilizing in-vivo and in-vitro pulldown assays followed by enzyme-coupled assays, the researchers identified the key segments of both Tma23 and Pxr1 that interact with Prp43 to modulate ATPase activity. Through cryo-EM studies and mutational assays, the authors characterized the structural interactions between Prp43 and both activators and concluded that the activating and inhibitory patches exhibited distinct binding patterns with Prp43. This data was further validated through HDX-MS and XL-MS analyses, which highlighted specific interactions across the RecA1 and RecA2 domains of Prp43 that corroborated the structural interactions derived from the cryo-EM analysis and alluded to potential allosteric effects. Lastly, using in-vivo mutational and growth analyses, the researchers evaluated the role of homodimerization of both Tma23 and Pxr1 in their interactions with Prp43. The authors' work is notable in that they identified a novel Prp43 G-patch activator, Tma23, and specifically characterized the binding interfaces between Tma23/Pxr1 and Prp43. Thus, the authors concluded the presence of allosteric inhibition that modulated the binding of Prp43 to its RNA substrate and its subsequent enzymatic activity. **The paper is overall very well written, and the authors' conclusions are generally supported by their empirical results and the biochemical context of the Prp43 system provided throughout the paper.**

We thank the Reviewer for his/her supportive comments.

Recommendations for revisions

Content

Page 10, "Interaction of Prp43 with Pxr1^{GM} and Tma23^{GM} revealed by HDX-MS analyses": **I'm curious why HDX-MS data was only shown for Prp43 and not for peptides belonging to Tma23 or Pxr1?**

Pxr1^{GM} and Tma23^{GM} form stoichiometric complexes with Prp43 when co-expressed in *E. coli* (Prp43:Pxr1^{GM} & Prp43:Tma23^{GM}). These complexes were used for cryo-EM, HDX-MS, XL-MS, and ATPase assays. Pxr1^M / Pxr1^{GM} and Tma23^M / Tma23^{GM} fragments expressed in *E. coli* are degradation prone, hence we were unable to perform HDX-MS analyses of the individual fragments. We have included a statement clarifying this (pages 5, 7).

Could HDX results validate the 43- or 13-residue segments the authors identified previously as I-patches for each activator?

The cryo-EM structure of the Prp43:Pxr1^{GM} complex (Figure 5a) shows how I-patch present within the 13-residue fragment of Pxr1 (P^{M13}; Figure 4k) binds to Prp43. Both T^{M43} and P^{M13} fragments which inhibit Prp43-ATPase activity contain the conserved FVKGE motif of the I-patch (Figure 4k) which is well-resolved in the cryo-EM structure.

We were unable to perform cryo-EM analyses of the Prp43:Tma23^{GM} since the complex dissociated during grid preparation. We resorted to HDX-MS, XL-MS and AlphaFold2 based modelling to investigate the organization and the dynamics of the Prp43:Tma23^{GM} complex. As a control for these studies, we have included the Prp43:Pxr1^{GM} complex. We find protection to HD exchange in the region of Prp43 which recruits the I-patch from Tma23 (region within T^{M43}) and Pxr1 (P^{M13}, control) (Figure 7a). A combination of cryo-EM, AlphaFold2 and HDX-MS thus provided insights into how an I-patch binds to Prp43 to inhibit its ATPase activity.

Could this allow the G-patches of each activator to be more precisely identified despite the lack of cryo-EM data?

Although we were unable to resolve the G-patches of Tma23 and Pxr1 on Prp43 through cryo-EM studies, HDX-MS studies of the Prp43:Tma23^{GM} and Prp43:Pxr1^{GM} complexes trace HD exchange protection in the immediate vicinity where the G-patch binds to Prp43 (Figure 7a, comparison with the DHX15:NKRF^G structure). The ability of G-patches within Tma23 and Pxr1 to activate Prp43 was demonstrated by employing ATPase assays (Figure 4a-d).

Page 12, “The C-terminal segments organize Pxr1 and Tma23 homodimers”: This section reads as out of place with respect to the remainder of the results section. Compared to their in-vivo, cryo-EM, and HDX-MS studies, this analysis on the homodimerization of Tma23 and Pxr1 does not seem as novel or significant. Besides identifying the dimerization interface of each protein, I believe further research into the structural details of this interaction, possibly using HDX-MS, cryo-EM, and other techniques used previously, would convey more importance to their dimerization analysis. Alternatively, the authors could expand on the biological significance of dimerization and further emphasize future directions of research.

A combination of yeast two-hybrid (Y2H), biochemical, genetic, and cell-biological data indicate that the I-patch and the dimerization module enable interactions with Prp43.

Y2H and biochemical studies: Full-length Pxr1^{F126A} interacts with Prp43, however, this interaction is lost when the dimerization module is absent (Pxr1^{M-F126A}) (Figure 6a & 6b, left panels). This is also the case for the Tma23^{F96A} mutant (Figure 6a & 6b, right panels).

Genetic studies: The individual Pxr1^{F96A} and the Pxr1^{GM}-Cov2^C mutants are not severely growth impaired like the Pxr1 depletion strain (Figure 6c). However, the Pxr1^{GM-F126A}-Cov2^C double mutant shows severe synergistic growth defects (Figure 6c). This growth defect was rescued by replacing the heterologous Cov2^C-dimerization by Tma23^D (Pxr1^{GM-F126A}-Tma23^D) (Figure 6d, left panel). A similar set of genetic interactions were observed for the Tma23^{F96A} mutant (Figure 6c & 6d, right panels).

Cell-biological studies: The synergistic interactions correlate with 60S export assays (Figure 6e). For e.g., the Pxr1^{GM-F126A}-Cov2^C mutant is severely impaired in 60S export as compared to the individual Pxr1^{GM-F126A} and Pxr1-Cov2^C mutants (Figure 6e). This defect is partially rescued in the Pxr1^{GM-F126A}-Tma23^D mutant (Figure 6e).

Altogether, the Y2H data, the cell-biological and the genetic data support the idea that the I-patch and the dimerization module, together, enable interactions with Prp43 during 60S assembly (page 13, 14 & Figure 6e).

While cryo-EM studies have revealed the basis underlying Prp43:I-patch interactions, it is unclear how the dimerization module enables interactions with Prp43. Full-length Tma23 and Pxr1 even in the presence of Prp43 expressed poorly in *E. coli*, therefore, we were unable to assemble these complexes. Determining the structures of native dimeric Tma23:Prp43 and Pxr1:Prp43 complexes is a future challenge.

CRAC studies showed that Prp43 crosslinks with multiple sites that are positioned as “pairs” within the 25S regions of the pre-rRNA (PMID: **19941813**; PMID: **19941819**). Our data provide support to the notion that dimeric Prp43 complexes operate during 60S assembly. However, which “pairs” are targeted by Tma23 and Pxr1 remains unclear. These challenges are discussed in the text (page 17).

Figures

Figure 2a: There needs to be more details provided to explain the system the authors are using for

the yeast two-hybrid assays. Consider adding a diagram for the pLexA and pAct2.2 vectors or a more detailed methods section on this system to further support the results. This inclusion would be helpful for understanding the use of both SD-Leu-Trp and SD-His plates, as well as offering more support for the observed interactions between Prp43 and Tma23/Pxr1.

We provide an explanation of the yeast two-hybrid system in the Methods Section (page 28).

Figures 2c-2f: Please include an explanation of the y-axis of the ATP hydrolysis figures, similar to the explanation given in the methods section, and how they relate to the reduction in A340nm as detailed in the text.

Done.

Figure 3a: Please explain the importance of temperature as a variable in the growth assays.

This is standard practice to investigate growth rates of yeast mutants at different temperatures to evaluate any temperature/cold sensitive phenotypes.

Figure 4b: Is the change in ATPase activity between Prp43-TG + TM43 and Prp43-TG + TM statistically significant? How can the 43-residue region, which the authors claim is the key inhibitory region of Tma23, show comparable if not slightly elevated levels of ATPase activity compared to Prp43-TG + TM?

T^M comprises the entire M domain within Tma23, whereas T^{M43} comprises a shorter 43 residue segment within T^M . This is indicated in Figure 4. We aimed to determine the minimal region within T^M that inhibits Prp43- T^G fusion ATPase activity. We therefore compared the ability of either T^M or the shorter T^{M43} to inhibit Prp43- T^G ATPase activity.

T^M dampens Prp43- T^G ATPase activity (from 80 $[E]^{-1}min^{-1}$ to 17.9 $[E]^{-1}min^{-1}$) by 4.4-fold (Figure 4b, compare column 3 & 4). The shorter version, T^{M43} , dampens Prp43- T^G ATPase activity (from 80 $[E]^{-1}min^{-1}$ to 21.6 $[E]^{-1}min^{-1}$) by 3.7-fold (Figure 4b, compare column 3 & 5). The slightly less inhibition (3.7-fold as compared to 4.4-fold) of Prp43- T^G ATPase activity by T^{M43} in comparison to T^M could be due to the missing N-terminal 24 residues. On the other hand, the entire M domain of Pxr1 (P^M) and a shorter 13-residue segment within P^M (P^{M13}) dampens Prp43- T^G ATPase activity by 2.9 and 3.2-fold, respectively (Figure 4d, compare columns 3 & 4; columns 3 & 5).

Figure 4d: The authors' identification of the minimal inhibitory region of Pxr1M13 could be better supported by comparing it to the reduction in ATPase activity of Prp43-TG + PM, similar to that shown in figure 4b for Prp43-TG + TM43 and Prp43-TG + TM.

We have added the data set into Figure 4d (column 4).

Figure 7b: Please specify either in the figure or the legend that the uptake curves are for peptides belonging to Prp43 and not the interacting activators.

We have specified in the Legend that all peptides correspond to Prp43 (page 22).

Perhaps move supplementary figure 5c to this figure? Additionally, consider specifying further in which regions (shown in 7a) each set of peptides belongs to.

The peptides used for the analyses along the Prp43 primary sequence in the presence and absence (-/+) Tma23^{GM}/Pxr1^{GM} are depicted in Figure 7b. We specify the peptide above each representative uptake curve (Figure 7c).

It's never explained which structure corresponds to cs.1 and cs.2, making it very confusing to connect the uptake curves to the structures provided.

"cs" refers to "charge state" of the identified peptide during mass spectrometry that is used for the analyses (Supplementary Table 2). This is now stated in the Figure Legend.

Supplementary figure 1d, 1e: Please add pLDDT scores for the provided AlphaFold models to support the use of these models later in the paper. I'm skeptical of AlphaFold's accuracy in modeling the loop regions and so it needs to be quantified.

We present pLDDT (per residue local confidence) and PAE (Predicted Aligned Error) analyses for all AlphaFold2 models presented in Supplementary Figure 1.

Supplementary figure 4: There are several instances throughout the paper (pgs. 9, 11) that the structure of Prp43-Tma23G/Pxr1G is compared to DHX15-NKRF, and further conclusions were made based on these similarities. This is the only figure that really addresses these similarities, although I don't believe it offers sufficient evidence to warrant the claims made regarding Tma23G/Pxr1G. Personally, I find this figure and its legend difficult to comprehend due to the variety of RNA helicases mentioned. Perhaps consider editing the figure to more specifically compare the structure of DHX15-NKRF to Prp43-Tma23G/Pxr1G.

The ability of the G-patch within Tma23 to activate Prp43 was demonstrated by employing ATPase assays (Figure 4a-d). We have employed Pxr1^G as a control, which is an established G-patch containing activator of Prp43 (PMID: 24823796). Based on the enzymatic assays, we conclude that Tma23^G activates Prp43.

How do Tma23^G and Pxr1^G interact with Prp43? We have compared Alphafold2 modelled Prp43:Tma23^G and Prp43:Pxr1^G complexes to the DHX15-NKRF^G crystal structure (Figure 7a, and Supplementary Figure 5b) (PMID: 32179686). DHX15 is the human homolog of yeast Prp43, whereas NKRF is the human functional equivalent of the yeast G-patch factor Pfa1. HDX-MS studies of the Prp43:Pxr1^{GM} & Prp43:Tma23^{GM} trace a H/D exchange protection path in the immediate vicinity where the G-patch of NKRF binds to DHX15 (Figure 7a).

Prp43/DHX15 and Prp2 are two known RNA helicases that are activated by different G-patch containing factors. The point of Supplementary Figure 4c is to place our cryo-EM structure of the Prp43:Pxr1^{GM} complex in context of numerous known Prp43/DHX15 and Prp2 conformational states (fully open to fully closed). The DHX15-NKRF^G structure is a fully closed state, whereas the Prp43:Pxr1^{GM} complex is in an open state.

Supplementary figure 5c: With the exception of the region 481-491, all of the regions shown in the uptake curves in figure 7 are shown to have relatively low or missing coverage in this figure. This makes me somewhat skeptical of the authors' HDX analysis. Furthermore, it is unclear as to whether this coverage map represents Prp43 bound to Tma23 or Pxr1. The authors state in the legend that it pertains to both bound activators but having two coverage maps for each activator would be a more effective supplement to the data in figure 7.

Thank you for pointing this out. We noticed that there was an error in the numbering of residues of Prp43 (Supplementary 5c) due to the presence of the tag. This has been rectified.

The HDX-MS derived exchange rates for peptide derived from Prp43 alone and when in complex with either Tma23^{GM} or Pxr1^{GM} is shown in Figure 7b. Representative uptake plots of specific peptides is shown in Figure 7c. The precise peptides used for these analyses are shown Supplementary Figure 5c.

A single protein (Prp43) coverage map is shown in Supplementary Figure 5c, as it is the same for all experiments (Prp43, Prp43:Tma23^{GM} & Prp43:Pxr1^{GM}). It would not be possible to compare protein HD exchange rates between Prp43 alone and Prp43 bound to Tma23^{GM} / Pxr1^{GM} if different peptides were used for the analyses.

Reviewer #3 (Remarks to the Author):

Portugal-Calisto and colleagues describe a new G-patch-containing factor, Tma23, that plays a role in the modulation of Prp43 activity during ribosome biogenesis. In addition to validating the Tma23 G-patch, their investigation of Tma23 function also uncovered a new structural module, conserved in both Tma23 and another Prp43 modulator, Pxr1, that negatively impacts Prp43 activity. Structural studies of this “I-patch” in complex with Prp43 suggest that it stabilizes an ADP-bound, “open” conformation of Prp43. Prp43 is involved in a multitude of RNA remodeling functions and understanding how its motor function is specifically targeted and regulated is of broad interest in multiple RNP assembly and catalytic processes. The identification and characterization of new Prp43 regulatory elements is therefore an important finding that merits publication in Nature Communications for a revised manuscript.

We thank the Reviewer for his/her constructive comments which have strengthened our conclusions.

There are a few issues that I would like the authors to address:

1. While using non-RNA stimulated ATPase activity is a useful initial assay to define the relative effects of G-patch and I-patch on Prp43 activity, this does not represent the biologically relevant catalytic activity of Prp43, which is translocation along ssRNA. The authors should address why they did not measure the effect of the I-patch (and G-patch) modules on RNA-stimulated ATPase activity. Their high throughput coupled ATPase assay is ideally suited to obtain curves showing the effect of increasing RNA concentration on the activity of Prp43 in the presence of various Tma23 and Pxr1 fragments. These results would not only define the effect of the I-patch in the proper mechanistic context but would also test if the presence of RNA substrate directly promotes I-patch release and subsequent G-patch binding.

We compared the ATPase activities of Prp43 alone / Prp43-T^G fusion when bound to Tma23^M / Pxr1^M, and in the absence and presence (-/+) of U₂₀ RNA (U₂₀).

U₂₀ stimulates the ATPase activity of Prp43 (grey) and the Prp43-T^G fusion (red) (Figure 4e & 4f; -/+ RNA columns).

However, the dampened ATPase activity for Prp43:Tma23^M, and Prp43-T^G:Tma23^M complexes was not reversed by the addition U₂₀ (Figure 4e & 4f; -/+ RNA columns). Similar observations were also made for Prp43, Prp43:Pxr1^M and Prp43-T^G:Pxr1^M complexes (Figure 4g & 4h; -/+ RNA columns). Fluorescence anisotropy-based measurements show that the affinity of Prp43 towards Cy5-labelled U₁₂ RNA was not altered when bound to either Tma23^M or Pxr1^M (Figure 4i).

All these studies show that RNA binding is unable reverse the inhibitory hold of the I-patch on Prp43-ATPase activity.

2. Along the same lines, why did the authors only measure V_{max} rather than obtaining full kinetic parameters for their Prp43 ATPase assays? Obtaining K_M's of Prp43 ATPase activity with various Tma23 and Pxr1 fragments would allow them to test the hypothesis that the I-patch-engaged state stabilizes the ADP-bound conformation of Prp43.

We obtained kinetic parameters for the ATPase activities of Prp43, Prp43:Tma23^M and Prp43:Pxr1^M complexes, in the presence and the absence of U₂₀. In the presence of U₂₀, we find that the K_M's for Prp43, Prp43:Tma23^M and Prp43:Pxr1^M are in a similar μM range, whereas the k_{cat} values show a strong decrease (Figure 4j). These data are consistent with a non-competitive mode of inhibition. We infer that the affinity of ATP toward Prp43 is not significantly affected when bound to Tma23^M / Pxr1^M. We speculate that I-patch traps the RecA domains into a conformation(s) that disfavours ATP hydrolysis. A definitive answer necessitates smFRET analyses that quantitatively monitors the different conformational states visited by Prp43 in the presence and absence of I-patch. Such

experiments performed by the Dr. Sarah Adio's group, Göttingen were critical to gain insights into the precise mechanism by which the G-patch activates Prp43 (PMID: 36409901; PMID: 37167006).

3. While the use of AlphaFold to model G-patch interactions seems reasonable, the use of AlphaFold to propose a model for Prp43-Tma23^{GM} in Figure 7 seems misguided. The authors show ample evidence that G-patch and I-patch interactions are likely exclusive (i.e. I-patch binds and stabilizes an inactive (ADP), open conformation while the G-patch stimulates the active and closed RNA-bound state). Yet in Figure 7, the AlphaFold model shows the Tma23 I-patch bound to the active, closed state and seems to suggest that G-patch and I-patch interactions can occur simultaneously. My guess is that AlphaFold is biased toward the more compact closed state and cannot predict the conformation stabilized by I-patch binding (The authors could test this by checking if AlphaFold accurately predicts the Prp43/Pxr1^M structure, including the RecA1/A2 orientation observed in the cryo-EM structure). More traditional homology modeling is likely a much better strategy to visualize the HDX-MS and XL-MS data, using separate models (and conformations) to model Tma23 G-patch and I-patch interactions with Prp43.

Thank you for raising this point. The “closed state” conformation of Prp43 within the Prp43:Tma23^G and Prp43:Pxr1^G complexes has now been modelled using AlphaFold2 (Figure 7a) to visualize the HDX- data. We have used the Prp43 “open state” conformation obtained from our Prp43:Pxr1^{GM} cryo-EM structure to visualize the HDX-MS data of how the I-patch of Tma23^{GM} docks onto Prp43 (Figure 7a).

As the Reviewer correctly notes, AlphaFold2 (also AlphaFold3) is biased, and models Prp43 into an RNA bound “closed state”. Yet, the algorithms seem to reliably locate I-patch on Prp43 (Figure 7a). We have used this model to visualize the XL-MS data that are restricted only to the immediate vicinity of the I-patch region (Figure 7d).

4. Figure 6A: I am concerned that the positive 2-hybrid signal for DBD-Pxr1^{F126A} x Prp43-AD and Tma23^{F96A} x Prp43-AD is being mediated by a “bridging” wild type copy of Pxr1 or Tma23. The only way I can think of controlling for this is to use Pxr1^{GM}-Cov2^C and Tma23^{GM}-Cov2^C as the “wild-type” genomic copy. The authors need to address this if they want to claim that the D-domain helps with Prp43 binding from 2-hybrid data alone.

The claim that the D-domain contributes to Prp43 binding is based on a combination of observations: Y2H, biochemistry, genetic and cell-biology.

Pxr1^{F126A} interacts with Prp43 (positive Y2H signal), but this interaction is strongly impaired when the dimerization module is missing (Pxr1^{M-F126A}, Figure 6a, rows 2, 3 & 4; Figure 6b, lanes 1 & 2).

This is also the case for the Tma23^{F96A} mutant (Figure 6a, rows 6,7 & 8; Figure 6b, lanes 4 & 5).

The individual Pxr1^{F96A} and the Pxr1^{GM}-Cov2^C mutants are not severely growth impaired like the Pxr1 depletion strain (Figure 6c). However, the Pxr1^{GM-F126A}-Cov2^C mutant shows severe growth defects like the Pxr1 depletion strain (Figure 6c, compare rows 1 & 5). This growth defect is partially rescued by replacing the heterologous Cov2^C dimerization by Tma23^D (Pxr1^{GM-F126A}-Tma23^D) (Figure 6d, compare rows 1 & 6). A similar set of genetic interactions were observed for the Tma23^{F96A} mutant (Figure 6c & 6d).

Our genetic studies correlate well with the 60S export assays. For e.g., while the Pxr1^{GM-F126A} and Pxr1-Cov2^C mutants are modestly impaired in 60S export, the Pxr1^{GM-F126A}-Cov2^C mutant shows severe impairment (Figure 6e). This synergistic 60S export defect is partially rescued in the Pxr1^{GM-F126A}-Tma23^D mutant (Figure 6e). Altogether, these data support the idea that the I-patch and the dimerization module of Tma23 and Pxr1 enable binding to Prp43 in vivo.

To support our claim, as suggested, we replaced the endogenous Pxr1^D with CoV2^C in the Y2H reporter strain (NMY32). In this modified strain (Figure 6a, Pxr1^{GM}-CoV2^C; right panel), we find that full-length Pxr1^{F126A} interacts with Ppr43 (Figure 6a, rows 9 & 10).

Attempts to replace endogenous Tma23^D with CoV2^C in the NMY32 strain were unsuccessful. Hence, we tested interactions between the functional Tma23^{GM}-Pxr1^D chimera and the mutant Tma23^{GM-F96A}-Pxr1^D with Prp43 in the modified strain (NMY32 Pxr1^{GM}-CoV2^C). We find that both constructs interact with Prp43 (Fig. 6a, rows 11 & 12). The interaction of Tma23^{GM-F96A}-Pxr1^D fusion with Prp43 is modestly impaired.

Minor issues:

1. Figure 2 – In the ATPase assays, there is no control for background/contaminating ATPase activity for Tma23^G, Pxr1^G and Pfa1^G in the absence of Prp43. I am assuming this background was measured and subtracted, but this is not explicitly stated in the methods. The small Pxr1^G stimulatory effect observed in the experiments makes this more important than usual.

We have employed Pxr1^G as a control for our analyses, which is an established G-patch activator of Prp43 (PMID: 24823796).

Prior to performing enzymatic assays, we have assessed contaminating ATPase activity in our preparations. Shown below (next page) are two examples: Prp43-ATPase activity in the presence of equimolar amounts of Tma23^G / Pxr1^G and U₂₀ RNA. The “initial reaction rates” for U₂₀ and 1XTma23^G / 1XPxr1^G (in the absence of Prp43) are negligible and are like the “buffer” control. Only those preparations which showed “initial reaction rates”, like the “buffer” control were used for the ATPase assays. We have included a statement clarifying this in the Methods (page 23).

B (light grey): Buffer contribution; B+U₂₀ (dark grey): Buffer + RNA contribution; G (brown): 1XTma23^G (left panel) / 1XPxr1^G (right panel) contribution.

2. Figure 3D – Why is this experiment carried out at 37°, when the growth defect associated with Tma23 depletion is much more pronounced at 30° (Fig. 3A)? It would be easier to define if Tma23L7E has an intermediate phenotype at the lower temp.

The P_{GAL1}-TMA23 strain when transformed with the Tma23^{L7E} mutant at 25 and 30°C gave rise to spontaneous suppressors, for unknown reasons. However, this was not the case at 37°C. Hence, we have provided the growth analyses for the Tma23^{L7E} expressing strain at 37°C. Curiously, the P_{GAL1}-PXR1 strain did not have any of these issues.

3. Last paragraph of “An inhibitory patch within...activity”/Figure 4D: The authors should discuss this result more, as it suggests that the effects of the G-patch and the I-patch addition are independent.

Was the reciprocal mix and match experiment also performed (i.e. Prp43-PG + Tma23M43)?

The aim of the ATPase assays with the Prp43-T^G chimera (Prp43-T^G:Tma23^M & Prp43-T^G:Pxr1^M in Figure 4b & 4c) was to provide an alternative definitive evidence that Tma23^M and Pxr1^M inhibit Prp43-ATPase activity. Prp43-T^G showed a strong ~30-fold stimulation of ATPase activity (Figure 2e) as compared to the Prp43-P^G fusion (~2-fold). Hence, the Prp43-T^G fusion was selected to test the inhibition by Tma23^M and Pxr1^M. The reciprocal experiment was not performed.

4. Cryo-EM sample preparation: Please include a more detailed description of how the sample used for cryo-EM was prepared, especially the relative concentrations of Prp43, Prx1^{GM} and nucleotide and the order of addition of the sample components.

Recombinant Tma23^{GM} and Tma23^M when expressed as individual fragments in *E. coli* are aggregation prone. However, they form stable soluble stoichiometric complexes Prp43:Pxr1^{GM} and Prp43:Tma23^{GM} when co-expressed with Prp43. These were used for cryo-EM sample preparation, HDX-MS analyses, and XL-MS analyses. A description of how the Prp43:Pxr1^{GM} complex containing grids were vitrified for cryo-EM studies is provided in the Methods Section (page 27).

5. Figure 5: In panel A, please show the actual cryo-EM density (low pass-filtered if needed to show all regions) next to the cartoon model.

Done.

Panel B – this is a minor point, but the arrangement of these panels (1,2,3) is in the opposite direction of the same regions in the inset of panel A (3,2,1 going right to left).

We have altered the arrangement of the Panel B to match the insets in Panel A to avoid confusion. It might be useful to label F126 and L132 in the panel A inset to help with navigation. Also, in panel B2, the presence of ADP is misleading and should be omitted. The perspective makes it look like the nucleotide is quite close when in fact it is probably 15Å away.

Done.

Figure 5C: In the text, the authors state that the “connecting loop moves toward...”. I think it would be more accurate to say that the loop adopts a different conformation due to the reorientation of the RecA domains. Even in the figure, it is obvious that the first strand in RecA2 moves as much as the loop.

Done (page 10).

Sup. Fig. 4B. It might be confusing to some readers that both nucleotides are labeled ADP, since one is really ADP-BeF3. Please alter in some way to make it clear that one pocket contains ADP and the other “ATP”.

Done.

6. Supplementary Figure 3: Please add a panel with experimental maps around the Pxr1 peptide as an example of the map quality.

Done.